# SGFormer: Simplifying and Empowering Transformers for Large-Graph Representations

**Qitian Wu**[1], **Wentao Zhao**[1], **Chenxiao Yang**[1], **Hengrui Zhang**[2], **Fan Nie**[1],
**Haitian Jiang**[3], **Yatao Bian**[4], **Junchi Yan**[1]*

[1] Dept. of Computer Science and Engineering & MoE Key Lab of AI, Shanghai Jiao Tong University
[2] Dept. of Computer Science, University of Illinois at Chicago
[3] Courant Institute, New York University
[4] Tencent AI Lab
{echo740,permanent,chr26195,youluo2001,yanjunchi}@sjtu.edu.cn,
hzhan55@uic.edu, hatian.jiang@nyu.edu, yatao.bian@gmail.com

## Abstract

Learning representations on large-sized graphs is a long-standing challenge due to the inter-dependence nature involved in massive data points. Transformers, as an emerging class of foundation encoders for graph-structured data, have shown promising performance on small graphs due to its global attention capable of capturing all-pair influence beyond neighboring nodes. Even so, existing approaches tend to inherit the spirit of Transformers in language and vision tasks, and embrace complicated models by stacking deep multi-head attentions. In this paper, we critically demonstrate that even using a *one-layer* attention can bring up surprisingly competitive performance across node property prediction benchmarks where node numbers range from thousand-level to billion-level. This encourages us to rethink the design philosophy for Transformers on large graphs, where the quadratic global attention is a computation overhead hindering the scalability. We frame the proposed scheme as Simplified Graph Transformers (SGFormer), which is empowered by a simple attention model that can efficiently propagate information among arbitrary nodes at the minimal cost of one propagation layer and linear complexity w.r.t. node numbers. Moreover, SGFormer requires none of positional encodings, feature/graph pre-processing or extra loss. Empirically, SGFormer successfully scales to the web-scale graph `ogbn-papers100M` and yields up to 141x inference acceleration over SOTA Transformers on medium-sized graphs. Beyond current results, we believe the proposed methodology alone enlightens a new technical path of independent interest for building Transformers on large graphs. The codes are publicly available at https://github.com/qitianwu/SGFormer.

## 1 Introduction

Learning on large graphs that connect interdependent data points is a fundamental problem in machine learning, with diverse applications in social and natural sciences [45; 15; 48; 43; 14]. One key challenge is to obtain effective node representations, especially under limited computation budget (e.g., time and space), that can be efficiently utilized for various downstream tasks.

Recently, Transformers have emerged as a popular class of foundation encoders for graph-structured data, by treating nodes in the graph as input tokens, and show highly competitive performance on

---

*Corresponding author. This work was partly supported by National Key Research and Development Program of China (2020AAA0107600), NSFC (62222607) and STCSM (22511105100).

37th Conference on Neural Information Processing Systems (NeurIPS 2023).

graph-level tasks [10; 66; 58; 63; 39] and node-level tasks [57; 54]. The global attention mechanism in Transformers [50] can capture implicit inter-dependencies among nodes that are not embodied by input graph structures, but could potentially make a difference in data generation (e.g., the undetermined structures of proteins that lack known tertiary structures [37; 64]). This advantage provides Transformers with the desired expressivity and leads to superior performance over graph neural networks in small-graph-based applications [64; 3; 11; 31; 36].

A concerning trend in current architectures is their tendency to automatically adopt the design philosophy of Transformers used in vision and language tasks [8; 2; 9]. This involves stacking deep multi-head attention layers, which results in large model sizes and a data-hungry nature. However, this design approach poses a significant challenge for Transformers to scale to large graphs.

• Due to the global all-pair attention mechanism, the time and space complexity of Transformers often scales quadratically with respect to the number of nodes, and the computation graph grows exponentially as the number of layers increases. Thereby, training deep Transformers for large graphs with node numbers in the millions can be extremely resource-intensive and may require delicate techniques for partitioning the inter-connected nodes into smaller mini-batches in order to mitigate computational overhead [57; 54; 35; 5].

• In small-graph-based tasks such as graph-level prediction for molecular property [18], where each instance is a graph and there are typically abundant labeled graph instances in a dataset, large Transformers may have sufficient supervision for generalization. However, in large-graph-based tasks such as node-level prediction for protein functions [19], where there is usually only a single graph and each node is an instance, labeled nodes may be relatively limited. This exacerbates the vulnerability of large Transformers to overfitting in such cases.

**Our Contributions.** This paper presents our initial attempt to reassess the need for deep attentions when learning representations on large graphs. Critically, we demonstrate that a *single-layer, single-head* attention model can perform surprisingly competitive across twelve graph benchmarks, with node numbers ranging from thousands to billions. Our model, framed as Simplified Graph Transformer (SGFormer), is equipped with a simple global attention that scales linearly w.r.t. the number of nodes. Despite its simplicity, this model retains the necessary expressivity to learn all-pair interactions without the need for any approximation. Furthermore, SGFormer does not require any positional encodings, feature or graph pre-processing, extra loss functions, or edge embeddings.

Experiments show that SGFormer achieves highly competitive performance in an extensive range of node property prediction datasets, which are used as common benchmarks for model evaluation w.r.t. the fundamental challenge of representation learning on graphs, when compared to powerful GNNs and state-of-the-art graph Transformers. Furthermore, SGFormer achieves up to 37x/141x speedup in terms of training/inference time costs over scalable Transformers on medium-sized graphs. Notably, SGFormer can scale smoothly to the web-scale graph `ogbn-papers100M` with 0.1B nodes, two orders-of-magnitude larger than the largest demonstration of existing works on graph Transformers.

Additionally, as a separate contribution, we provide insights into why the one-layer attention model can be a powerful learner for learning (node) representations on large graphs. Specifically, by connecting the Transformer layer to a well-established signal denoising problem with a particular optimization objective, we demonstrate that a one-layer attention model can produce the same denoising effect as multi-layer attentions. Furthermore, a single attention layer can contribute to a steepest descent on the optimization objective. These results suggest that a one-layer attention model is expressive enough to learn global information from all-pair interactions.

## 2   Preliminary and Related Work

We denote a graph as $\mathcal{G} = (\mathcal{V}, \mathcal{E})$ where the node set $\mathcal{V}$ comprises $N$ nodes, and the edge set $\mathcal{E} = \{(u, v) \mid a_{uv} = 1\}$ is defined by a symmetric (and usually sparse) adjacency matrix $\mathbf{A} = [a_{uv}]_{N \times N}$, where $a_{uv} = 1$ if nodes $u$ and $v$ are connected, and $0$ otherwise. Each node has an input feature $\mathbf{x}_u \in \mathbb{R}^D$ and a label $y_u$. The nodes in the graph are only partially labeled, forming a node set denoted as $\mathcal{V}_{tr} \subset \mathcal{V}$ (wherein $|\mathcal{V}_{tr}| \ll N$). Learning representations on graphs aims to produce node embeddings $\mathbf{z}_u \in \mathbb{R}^d$ that are useful for downstream tasks. The size of the graph, as measured by the number of nodes $N$, can be arbitrarily large, usually ranging from thousands to billions.

**Graph Neural Networks.** The layer-wise updating rule for GNNs [44; 23] can be defined as recursively aggregating the embeddings of neighboring nodes to compute the node representation:

$$\mathbf{z}_u^{(k+1)} = \eta^{(k)}(\bar{\mathbf{z}}_u^{(k+1)}), \quad \bar{\mathbf{z}}_u^{(k+1)} = \text{Agg}(\{\mathbf{z}_v^{(k)} | v \in \mathcal{R}(u)\}), \quad (1)$$

where $\mathbf{z}_u^{(k)}$ denotes the embedding at the $k$-th layer, $\eta$ denotes (parametric) feature transformation, and Agg is an aggregation function over the embeddings of nodes in $\mathcal{R}(u)$ which is the receptive field of node $u$ determined by $\mathcal{G}$. Common GNNs, such as GCN [23] and GAT [51], along with their numerous successors, typically assume that $\mathcal{R}(u)$ is the set of first-order neighboring nodes in $\mathcal{G}$. By stacking multiple layers (e.g., $K$) of the updating rule in (1), the model can integrate information from the local neighborhood into the representation. Because GNNs involve neighboring nodes in the computation graph, which exponentially increase w.r.t. $K$, training them on large graphs (e.g., with a million nodes) in a full-batch manner can be challenging. To reduce the overhead, GNNs require subtle techniques like neighbor sampling [65], graph partition [7], or historical embeddings [12]. Some works adopt knowledge distillation for inference on sparser graphs [61; 67] to accelerate inference, and more recently, MLP architectures are used to replace GNNs to accelerate training [60; 17].

**Graph Transformers.** Beyond message passing within local neighborhood, Transformers have recently gained attention as powerful graph encoders [10; 66; 63; 34; 25; 58; 4; 20; 39; 22]. These models leverage global all-pair attention, which aggregates all node embeddings to update the representation of each node. Global attention can be seen as a generalization of GNNs' message passing to a densely connected graph where $\mathcal{R}(u) = \mathcal{V}$, and equips the model with the ability to capture implicit dependencies such as long-range interactions or unobserved potential links in the graph. However, the all-pair attention incurs $O(N^2)$ complexity and becomes a computation bottleneck that limits most Transformers to handling only small-sized graphs (with up to hundreds of nodes). For larger graphs, recent efforts have resorted to strategies such as sampling a small (relative to $N$) subset of nodes for attention computation [16] or using ego-graph features as input tokens [68]. However, these strategies sacrifice the expressivity needed to capture all-pair interactions among arbitrary nodes. Another line of recent works design new attention mechanisms that can efficiently achieve all-pair message passing within linear complexity. One of the pioneering work [57] proposes kernelized Gumbel-Softmax message passing that utilizes random feature maps to approximate the all-pair attention with linear complexity. Another work [54] leverages the principled graph diffusion equation and resorts to the non-local diffusion operator to derive a linear attention network that models all-pair interactions without any approximation.

Another observation is that nearly all of the Transformer models mentioned above tend to stack deep multi-head attention layers, in line with the design of large models used in vision and language tasks [8; 2; 9]. However, this architecture presents challenges for scaling to industry-scale graphs, where $N$ can reach billions. Moreover, the model can become vulnerable to overfitting when the number of labeled nodes $|\mathcal{V}_{tr}|$ is much smaller than $N$. This is a common issue in extremely large graphs where node labels are scarce [19].

## 3 Simplifying and Empowering Transformers on Large Graphs

In this section, we introduce our approach, which we refer to as Simplified Graph Transformers (SGFormer). The key component of SGFormer is a simple global attention model that captures the implicit dependencies among nodes with linear complexity.

**Input Layer.** We first use a neural layer to map input features $\mathbf{X} = [\mathbf{x}_u]_{u=1}^N$ to node embeddings in the latent space, i.e., $\mathbf{Z}^{(0)} = f_I(\mathbf{X})$ where $f_I$ can be a shallow (e.g., one-layer) MLP. The node embeddings $\mathbf{Z}^{(0)} \in \mathbb{R}^{N \times d}$ will be used for subsequent attention computation and propagation.

**Simple Global Attention.** The global interactions in our model are captured by an all-pair attention unit that computes pair-wise influence between arbitrary node pairs. Unlike other Transformers, which often require multiple attention layers for desired capacity, we found that a single-layer global attention is sufficient. This is because through one-layer propagation over a densely connected attention graph, the information of each node can be adaptively propagated to arbitrary nodes within the batch. Therefore, despite its simplicity, our model has sufficient expressivity to capture implicit dependencies among arbitrary node pairs while significantly reducing computational overhead. In

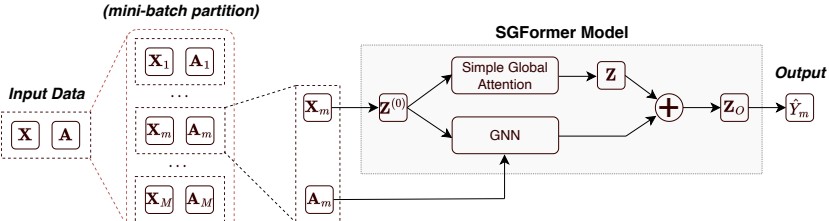

Figure 1: Illustration of the proposed model SGFormer and its data flow. The input graph data entails node features $\mathbf{X}$ and graph adjacency $\mathbf{A}$. For large graphs, we need to use mini-batch sampling that randomly partitions the input graph into mini-batches with smaller sizes. Each mini-batch is composed of the features of the nodes within this mini-batch $\mathbf{X}_m$ and the local graph adjacency $\mathbf{A}_m$ (one can also use neighbor sampling as an alternative). The mini-batch data $(\mathbf{X}_m, \mathbf{A}_m)$ (for large graphs) or the whole graph data $(\mathbf{X}, \mathbf{A})$ (for small graphs) will be fed into the SGFormer model that is implemented with a one-layer global attention and a GNN network. The model outputs the node representations for final prediction.

specific, we adopt a linear attention function defined as follows:

$$\mathbf{Q} = f_Q(\mathbf{Z}^{(0)}), \quad \tilde{\mathbf{Q}} = \frac{\mathbf{Q}}{\|\mathbf{Q}\|_{\mathcal{F}}}, \quad \mathbf{K} = f_K(\mathbf{Z}^{(0)}), \quad \tilde{\mathbf{K}} = \frac{\mathbf{K}}{\|\mathbf{K}\|_{\mathcal{F}}}, \quad \mathbf{V} = f_V(\mathbf{Z}^{(0)}), \qquad (2)$$

$$\mathbf{D} = \text{diag}\left(\mathbf{1} + \frac{1}{N}\tilde{\mathbf{Q}}(\tilde{\mathbf{K}}^{\top}\mathbf{1})\right), \quad \mathbf{Z} = \beta\mathbf{D}^{-1}\left[\mathbf{V} + \frac{1}{N}\tilde{\mathbf{Q}}(\tilde{\mathbf{K}}^{\top}\mathbf{V})\right] + (1-\beta)\mathbf{Z}^{(0)}, \qquad (3)$$

where $f_Q$, $f_K$, $f_V$ are all shallow neural layers (e.g., a linear feed-forward layer in our implementation), $\|\cdot\|$ denotes the Frobenius norm, $\mathbf{1}$ is an $N$-dimensional all-one column vector, the diag operation changes the $N$-dimensional column vector into a $N \times N$ diagonal matrix, and $\beta$ is a hyper-parameter for residual link. In Eqn. 3, $\mathbf{Z}$ combines the all-pair attentive propagation over $N$ nodes and the self-loop propagation. The former allows the model to capture the influence from other nodes, while the latter preserves the information of the centered nodes. The computation of Eqn. (3) can be achieved in $\mathcal{O}(N)$ time complexity, which is much more efficient than the Softmax attention in original Transformers [50] that requires $\mathcal{O}(N^2)$. While the Softmax attention possesses provable expressivity [1], its quadratic complexity hinders the scalability for large graphs. Our adopted design reduces the quadratic complexity to $\mathcal{O}(N)$ and in the meanwhile guarantee the expressivity for learning all-pair interactions. Therefore, it can successfully be applied for learning representations on graphs with a large number of nodes at the cost of moderate GPU memory.

**Incorporation of Structural Information.** For accommodating the prior information of the input graph $\mathcal{G}$, existing models tend to use positional encodings [39], edge regularization loss [57] or augmenting the Transformer layers with GNNs [58]. Here we resort to a simple-yet-effective scheme that combines $\mathbf{Z}$ with the propagated embeddings by GNNs at the output layer:

$$\mathbf{Z}_O = (1-\alpha)\mathbf{Z} + \alpha\text{GN}(\mathbf{Z}^{(0)}, \mathbf{A}), \quad \hat{Y} = f_O(\mathbf{Z}_O), \qquad (4)$$

where $\alpha$ is a weight hyper-parameter, and the GN module can be a simple GNN architecture (e.g., GCN [23]) that possesses good scalability for large graphs. The output function $f_O$ maps the final representation $\mathbf{Z}_O$ to prediction depending on specific downstream tasks, and in our implementation $f_O$ is a linear feed-forward layer.

**Complexity Analysis.** The overall computational complexity of our model is $\mathcal{O}(N + E)$, where $E = |\mathcal{E}|$, as the GN module requires $\mathcal{O}(E)$. Due to the typical sparsity of graphs (i.e., $E \ll N^2$), our model can scale linearly w.r.t. graph sizes. Furthermore, with only one-layer global attention and simple GNN architectures, our model is fairly lightweight, enabling efficient training and inference.

**Scaling to Larger Graphs.** For larger graphs that even GCN cannot be trained on using full-batch processing with a single GPU, we can use the random mini-batch partitioning method utilized by [57; 54]. This incurs only negligible additional costs during training and allows the model to scale to arbitrarily large graphs. Moreover, due to the linear complexity of our all-pair attention mechanism, we can employ large batch sizes, which facilitate the model in capturing informative global interactions among nodes within each mini-batch. Our model is also compatible with advanced techniques such as neighbor sampling [65], graph clustering [7], and historical embeddings [12], which we leave for future exploration along this orthogonal research direction. Fig. 1 presents the data flow of the proposed model.

Table 1: Comparison of (typical) graph Transformers w.r.t. required components (PE for positional embeddings, MA for multiple attention heads, AL for augmented training loss and EE for edge embeddings), all-pair expressivity and algorithmic complexity w.r.t. node number $N$ and edge number $E$ (often $E \ll N^2$). The largest demonstration means the largest graph size used by the papers.

| Model | Model Components | | | | All-pair Expressivity | Algorithmic Complexity | | Largest Demo. |
|---|---|---|---|---|---|---|---|---|
| | PE | MA | AL | EE | | Pre-processing | Training | |
| GraphTransformer [10] | R | R | - | R | Yes | $O(N^3)$ | $O(N^2)$ | 0.2K |
| Graphormer [63] | R | R | - | R | Yes | $O(N^3)$ | $O(N^2)$ | 0.3K |
| GraphTrans [58] | - | R | - | - | Yes | - | $O(N^2)$ | 0.3K |
| SAT [4] | R | R | - | - | Yes | $O(N^3)$ | $O(N^2)$ | 0.2K |
| EGT [20] | R | R | R | R | Yes | $O(N^3)$ | $O(N^2)$ | 0.5K |
| GraphGPS [39] | R | R | - | R | Yes | $O(N^3)$ | $O(N+E)$ | 1.0K |
| Gophormer [68] | R | R | R | - | No | - | $O(Nsm^2)$ | 20K |
| NodeFormer [57] | R | R | R | - | Yes | - | $O(N+E)$ | 2.0M |
| DIFFormer [54] | - | R | - | - | Yes | - | $O(N+E)$ | 1.6M |
| **SGFormer** (ours) | - | - | - | - | Yes | - | $O(N+E)$ | 0.1B |

# 4 Comparison with Existing Models

In this section, we provide a more in-depth discussion comparing our model with prior art and illuminating its potential in wide application scenarios. Table 1 presents a head-to-head comparison of current graph Transformers in terms of their architectures, expressivity, and scalability. Most existing Transformers have been developed and optimized for graph classification tasks on small graphs, while some recent works have focused on Transformers for node classification, where the challenge of scalability arises due to large graph sizes.

• **Architectures.** Regarding model architectures, some existing models incorporate edge/positional embeddings (e.g., Laplacian decomposition features [10], degree centrality [63], Weisfeiler-Lehman labeling [66]) or utilize augmented training loss (e.g., edge regularization [20; 57]) to capture graph information. However, the positional embeddings require an additional pre-processing procedure with a complexity of up to $O(N^3)$, which can be time- and memory-consuming for large graphs, while the augmented loss may complicate the optimization process. Moreover, existing models typically adopt a default design of stacking deep multi-head attention layers for competitive performance. In contrast, SGFormer does not require any of positional embeddings, augmented loss or pre-processing, and only uses a single-layer, single-head global attention, making it both efficient and lightweight.

• **Expressivity.** There are some recently proposed graph Transformers for node classification [68; 5] that limit the attention computation to a subset of nodes, such as neighboring nodes or sampled nodes from the graph. This approach allows linear scaling w.r.t. graph sizes, but sacrifices the expressivity for learning all-pair interactions. In contrast, NodeFormer [57] and SGFormer maintain attention computation over all $N$ nodes in each layer while still achieving $O(N)$ complexity. However, unlike NodeFormer which relies on random features for approximation, SGFormer does not require any approximation or stochastic components and is more stable during training.

• **Scalability.** In terms of algorithmic complexity, most existing graph Transformers have $\mathcal{O}(N^2)$ complexity due to global all-pair attention, which is a critical computational bottleneck that hinders their scalability even for medium-sized graphs with thousands of nodes. While neighbor sampling can serve as a plausible remedy, it often sacrifices performance due to the significantly reduced receptive field [35]. SGFormer scales linearly w.r.t. $N$ and supports full-batch training on large graphs with up to 0.1M nodes. For further larger graphs, SGFormer is compatible with mini-batch training using large batch sizes, which allows the model to capture informative global information while having a negligible impact on performance. Notably, due to the linear complexity and simple architecture, SGFormer can scale to the web-scale graph `ogbn-papers100M` (with 0.1B nodes) when trained on a single GPU, two orders-of-magnitude larger than the largest demonstration of NodeFormer [57] and DIFFormer [54].

# 5 Empirical Evaluation

We apply SGFormer to various datasets of node property prediction tasks, which are used as common benchmarks for evaluating the model's efficacy for learning effective representations and scalability to large graphs. In Sec. 5.1, we test SGFormer on medium-sized graphs (from 2K to 30K nodes) and compare it with an extensive set of expressive GNNs and Transformers. In Sec. 5.2, we scale SGFormer to large-sized graphs (from 0.1M to 0.1B nodes) where its superiority is demonstrated over

Table 2: Mean and standard deviation (with five independent runs using random initializations) of testing accuracy on medium-sized node classification benchmarks. We annotate the node and edge number of each dataset and OOM indicates out-of-memory when training on a GPU with 24GB memory. We highlight the best model with purple and the runner-up with brown in each dataset.

| Dataset | Cora | CiteSeer | PubMed | Actor | Squirrel | Chameleon | Deezer |
|---|---|---|---|---|---|---|---|
| # nodes | 2,708 | 3,327 | 19,717 | 7,600 | 2223 | 890 | 28,281 |
| # edges | 5,278 | 4,552 | 44,324 | 29,926 | 46,998 | 8,854 | 92,752 |
| GCN | 81.6 ± 0.4 | 71.6 ± 0.4 | 78.8 ± 0.6 | 30.1 ± 0.2 | 38.6 ± 1.8 | 41.3 ± 3.0 | 62.7 ± 0.7 |
| GAT | 83.0 ± 0.7 | 72.1 ± 1.1 | 79.0 ± 0.4 | 29.8 ± 0.6 | 35.6 ± 2.1 | 39.2 ± 3.1 | 61.7 ± 0.8 |
| SGC | 80.1 ± 0.2 | 71.9 ± 0.1 | 78.7 ± 0.1 | 27.0 ± 0.9 | 39.3 ± 2.3 | 39.0 ± 3.3 | 62.3 ± 0.4 |
| JKNet | 81.8 ± 0.5 | 70.7 ± 0.7 | 78.8 ± 0.7 | 30.8 ± 0.7 | 39.4 ± 1.6 | 39.4 ± 3.8 | 61.5 ± 0.4 |
| APPNP | 83.3 ± 0.5 | 71.8 ± 0.5 | 80.1 ± 0.2 | 31.3 ± 1.5 | 35.3 ± 1.9 | 38.4 ± 3.5 | 66.1 ± 0.6 |
| H$_2$GCN | 82.5 ± 0.8 | 71.4 ± 0.7 | 79.4 ± 0.4 | 34.4 ± 1.7 | 35.1 ± 1.2 | 38.1 ± 4.0 | 66.2 ± 0.8 |
| SIGN | 82.1 ± 0.3 | 72.4 ± 0.8 | 79.5 ± 0.5 | 36.5 ± 1.0 | 40.7 ± 2.5 | 41.7 ± 2.2 | 66.3 ± 0.3 |
| CPGNN | 80.8 ± 0.4 | 71.6 ± 0.4 | 78.5 ± 0.7 | 34.5 ± 0.7 | 38.9 ± 1.2 | 40.8 ± 2.0 | 65.8 ± 0.3 |
| GloGNN | 81.9 ± 0.4 | 72.1 ± 0.6 | 78.9 ± 0.4 | 36.4 ± 1.6 | 35.7 ± 1.3 | 40.2 ± 3.9 | 65.8 ± 0.8 |
| Graphormer$_{\text{SMALL}}$ | OOM | OOM | OOM | OOM | OOM | OOM | OOM |
| Graphormer$_{\text{SMALLER}}$ | 75.8 ± 1.1 | 65.6 ± 0.6 | OOM | OOM | 40.9 ± 2.5 | 41.9 ± 2.8 | OOM |
| Graphormer$_{\text{ULTRASMALL}}$ | 74.2 ± 0.9 | 63.6 ± 1.0 | OOM | 33.9 ± 1.4 | 39.9 ± 2.4 | 41.3 ± 2.8 | OOM |
| GraphTrans$_{\text{SMALL}}$ | 80.7 ± 0.9 | 69.5 ± 0.7 | OOM | 32.6 ± 0.7 | 41.0 ± 2.8 | 42.8 ± 3.3 | OOM |
| GraphTrans$_{\text{ULTRASMALL}}$ | 81.7 ± 0.6 | 70.2 ± 0.8 | 77.4 ± 0.5 | 32.1 ± 0.8 | 40.6 ± 2.4 | 42.2 ± 2.9 | OOM |
| NodeFormer | 82.2 ± 0.9 | 72.5 ± 1.1 | 79.9 ± 1.0 | 36.9 ± 1.0 | 38.5 ± 1.5 | 34.7 ± 4.1 | 66.4 ± 0.7 |
| **SGFormer** | **84.5 ± 0.8** | **72.6 ± 0.2** | **80.3 ± 0.6** | **37.9 ± 1.1** | **41.8 ± 2.2** | **44.9 ± 3.9** | **67.1 ± 1.1** |

scalable GNNs and Transformers. Later in Sec. 5.3 and 5.4, we compare the time/space efficiency and analyze the impact of several key components in our model, respectively. More detailed descriptions for datasets and implementation are deferred to Appendix B and C, respectively.

## 5.1 Results on Medium-sized Graphs

**Setup.** We first evaluate the model on commonly used graph datasets, including three citation networks `cora`, `citeseer` and `pubmed`, where the graphs have high homophily ratios, and four heterophilic graphs `actor`, `squirrel`, `chameleon` and `deezer-europe`, where neighboring nodes tend to have distinct labels. These graphs have 2K-30K nodes. For citation networks, we follow the semi-supervised setting of [23] for data splits. For `actor` and `deezer-europe`, we use the random splits of the benchmarking setting in [32]. For `squirrel` and `chameleon`, we use the splits proposed by a recent evaluation paper [38] that filters the overlapped nodes in the original datasets.

**Competitors.** Given the moderate sizes of graphs where most of existing models can scale smoothly, we compare with multiple sets of competitors from various aspects. Basically, we adopt standard GNNs including GCN [23], GAT [51] and SGC [52] as baselines. On top of this, we compare with advanced GNN models, including JKNet [59], APPNP [24], SIGN [40], H2GCN [72], CPGNN [71] and GloGNN [27]. In terms of Transformers, we mainly compare with the state-of-the-art models designed for node classification NodeFormer [57]. Furthermore, we adapt two powerful Transformers tailored for graph classification, i.e., Graphormer [63] and GraphTrans [58], for comparison. In particular, since the original implementations of these models are of large sizes and are difficult to scale on all node classification datasets considered in this paper, we adopt their smaller versions for our experiments. We use Graphormer$_{\text{SMALL}}$ (6 layers and 32 heads), Graphormer$_{\text{SMALLER}}$ (3 layers and 8 heads) and Graphormer$_{\text{ULTRASMALL}}$ (2 layers and 1 head). As for GraphTrans, we use GraphTrans$_{\text{SMALL}}$(3 layers and 4 heads) and GraphTrans$_{\text{ULTRASMALL}}$ (2 layers and 1 head).

**Results.** Table 2 reports the results of all the models. We found that SGFormer significantly outperforms three standard GNNs (GCN, GAT and SGC) by a large margin, with up to $25.9\%$ impv. over GCN on `actor`, which suggests that our one-layer global attention model is indeed effective despite its simplicity. Moreover, we observe that the relative improvements of SGFormer over three standard GNNs are overall more significant on heterophilic graphs `actor`, `squirrel`, `chameleon` and `deezer`. The possible reason is that in such cases the global attention could help to filter out spurious edges from neighboring nodes of different classes and accommodate dis-connected yet informative nodes in the graph. Compared to other advanced GNNs and graph Transformers (particularly NodeFormer), the performance of SGFormer is highly competitive and even superior with significant gains over the runner-ups in most cases. These results serve as concrete evidence for verifying the efficacy of SGFormer as a powerful learner for node-level prediction. We also found that both Graphormer and GraphTrans suffer from serious over-fitting, due to their relatively complex

Table 3: Testing results (ROC-AUC for `ogbn-proteins` and Accuracy for other datasets) on large-sized node property prediction benchmarks. The training of NodeFormer on `ogbn-papers100M` cannot be finished within an acceptable time budget.

| Method | ogbn-proteins | Amazon2m | pokec | ogbn-arxiv | ogbn-papers100M |
|---|---|---|---|---|---|
| **# nodes** | 132,534 | 2,449,029 | 1,632,803 | 169,343 | 111,059,956 |
| **# edges** | 39,561,252 | 61,859,140 | 30,622,564 | 1,166,243 | 1,615,685,872 |
| MLP | 72.04 ± 0.48 | 63.46 ± 0.10 | 60.15 ± 0.03 | 55.50 ± 0.23 | 47.24 ± 0.31 |
| GCN | 72.51 ± 0.35 | 83.90 ± 0.10 | 62.31 ± 1.13 | **71.74 ± 0.29** | OOM |
| SGC | 70.31 ± 0.23 | 81.21 ± 0.12 | 52.03 ± 0.84 | 67.79 ± 0.27 | 63.29 ± 0.19 |
| GCN-NSampler | 73.51 ± 1.31 | 83.84 ± 0.42 | 63.75 ± 0.77 | 68.50 ± 0.23 | 62.04 ± 0.27 |
| GAT-NSampler | 74.63 ± 1.24 | 85.17 ± 0.32 | 62.32 ± 0.65 | 67.63 ± 0.23 | 63.47 ± 0.39 |
| SIGN | 71.24 ± 0.46 | 80.98 ± 0.31 | 68.01 ± 0.25 | 70.28 ± 0.25 | **65.11 ± 0.14** |
| NodeFormer | **77.45 ± 1.15** | **87.85 ± 0.24** | **70.32 ± 0.45** | 59.90 ± 0.42 | - |
| **SGFormer** | **79.53 ± 0.38** | **89.09 ± 0.10** | **73.76 ± 0.24** | **72.63 ± 0.13** | **66.01 ± 0.37** |

Table 4: Efficiency comparison of SGFormer and graph Transformer competitors w.r.t. training time (ms) per epoch, inference time (ms) and GPU memory (GB) costs on a Tesla T4. We use the small model versions of Graphormer and GraphTrans. The missing results are caused by out-of-memory.

| Method | Cora | | | PubMed | | | Amazon2M | | |
|---|---|---|---|---|---|---|---|---|---|
| | Tr (ms) | Inf (ms) | Mem (GB) | Tr (ms) | Inf (ms) | Mem (GB) | Tr (ms) | Inf (ms) | Mem (GB) |
| Graphormer | 563.5 | 537.1 | 5.0 | - | - | - | - | - | - |
| GraphTrans | 160.4 | 40.2 | 3.8 | - | - | - | - | - | - |
| NodeFormer | 68.5 | 30.2 | 1.2 | 321.4 | 135.5 | 2.9 | 5369.5 | 1410.0 | 4.6 |
| **SGFormer** | 15.0 | 3.8 | 0.9 | 15.4 | 4.4 | 1.0 | 2481.4 | 382.5 | 2.7 |

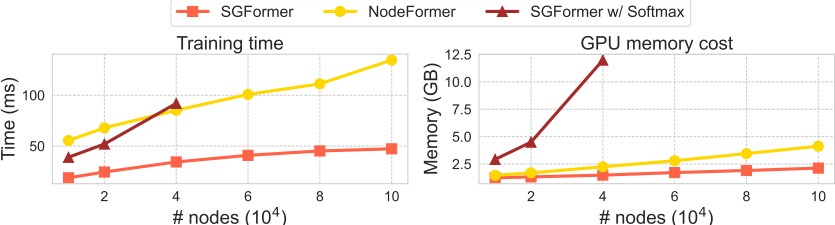

Figure 2: Scalability test of training time per epoch and GPU memory usage w.r.t. graph sizes (a.k.a. node numbers). NodeFormer suffers out-of-memory when # nodes reaches more than 30K.

architectures and limited ratios of labeled nodes. In contrast, the simple and lightweight architecture of SGFormer leads to its better generalization ability given the limited supervision in these datasets.

## 5.2 Results on Large-sized Graphs

**Setup.** We further evaluate the model on several large-graph datasets where the numbers of nodes range from millions to billions. The citation network `ogbn-arxiv` and the protein interaction network `ogbn-proteins` contain 0.16M and 0.13M nodes, respectively, and these two graphs have different edge sparsity. We use the public splits in OGB [19]. Furthermore, the item co-occurrence network `Amazon2M` and the social network `pokec` consist of 2.0M and 1.6M node, respectively, and the latter is a heterophilic graph. We follow the splits used in the recent work [57] for `Amazon2M` and use random splits with the ratio 1:1:8 for `pokec`. The largest dataset we demonstrate is `ogbn-papers100M` where the node number reaches 0.1B and we also follow the public OGB splits.

**Competitors.** Due to the large graph sizes, most of the expressive GNNs and Transformers compared in Sec. 5.1 are hard to scale within acceptable computation budgets. Therefore, we compare with MLP, GCN and two scalable GNNs, SGC and SIGN. We also compare with GNNs using the neighbor sampling [65]: GCN-NSampler and GAT-NSampler. Our main competitor is NodeFormer [57], the recently proposed scalable graph Transformer with all-pair attention.

**Results.** Table 3 presents the experimental results. We found that SGFormer yields consistently superior results across five datasets, with significant performance improvements over GNN competitors. This suggests the effectiveness of the global attention that can learn implicit inter-dependencies among a large number of nodes beyond input structures. Furthermore, SGFormer outperforms NodeFormer by a clear margin across all the cases, which demonstrates the superiority of SGFormer that uses simpler architecture and achieves better performance on large graphs. For the largest dataset `ogbn-papers100M` where prior Transformer models fail to demonstrate, SGFormer scales smoothly

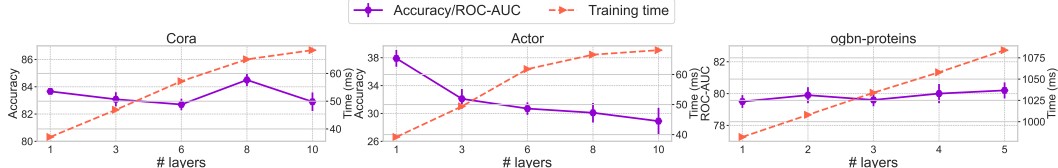

Figure 3: Testing scores and training time per epoch of SGFormer w.r.t. # attention layers. More results on more datasets are deferred to Appendix D.

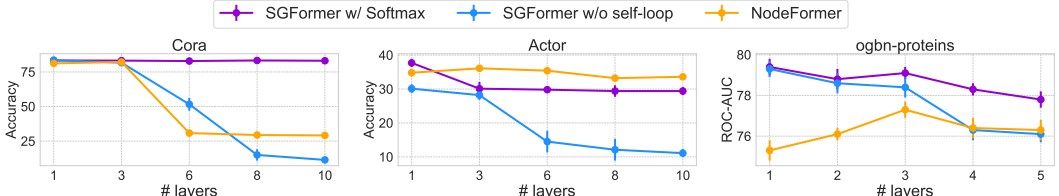

Figure 4: Testing performance of NodeFormer, SGFormer w/o self-loop and SGFormer w/ Softmax w.r.t. the number of attention layers. The missing results are caused by out-of-memory.

with decent efficiency and yields highly competitive results. Specifically, SGFormer reaches the testing accuracy of 66.0 with consumption of about 3.5 hours and 23.0 GB memory on a single GPU for training. This result provides strong evidence that shows the promising power of SGFormer on extremely large graphs, producing superior performance with limited computation budget.

## 5.3 Efficiency and Scalability

We next provide more quantitative comparisons w.r.t. the efficiency and scalability of SGFormer with the graph Transformer competitors Graphormer, GraphTrans, and NodeFormer.

**Efficiency Comparison.** Table 4 reports the training time per epoch, inference time and GPU memory costs on `cora`, `pubmed` and `Amazon2M`. Since the common practice for model training in these datasets is to use a fixed number of training epochs, we report the training time per epoch here for comparing the training efficiency. We found that notably, SGFormer is orders-of-magnitude faster than other competitors. Compared to Graphormer and GraphTrans that require quadratic complexity for global attention and are difficult for scaling to graphs with thousands of nodes, SGFormer significantly reduces the memory costs due to the simple global attention of $O(N)$ complexity. In terms of time costs, SGFormer yields 38x/141x training/inference speedup over Graphormer on `cora`, and 20x/30x training/inference speedup over NodeFormer on `pubmed`. This is mainly because Graphormer requires the compututation of quadratic global attention that is time-consuming, and NodeFormer additionally employs the augmented loss and Gumbel tricks. On the large graph `Amazon2M`, where NodeFormer and SGFormer both leverage mini-batch training, SGFormer is 2x and 4x faster than NodeFormer in terms of training and inference, respectively.

**Scalability Test.** We further test the model's scalability w.r.t. the numbers of nodes within one computation batch. We adopt the `Amazon2M` dataset and randomly sample a subset of nodes with the node number ranging from 10K to 100K. For fair comparison, we use the same hidden size 256 for all the models. In Fig. 2, we can see that the time and memory costs of SGFormer both scale linearly w.r.t. graph sizes. When the node number goes up to 40K, the model (SGFormer w/ Softmax) that replaces our attention with the Softmax attention suffers out-of-memory and in contrast, SGFormer costs only 1.5GB memory.

## 5.4 Further Discussions

We proceed to analyze the impact of model layers on the testing performance and further discuss the performance variation of other models w.r.t. the number of attention layers. Due to space limit, we defer more results to Appendix D.

**How does one-layer global attention perform compared to multi-layer attentions of Eqn. 5?** Fig. 3 presents the testing performance and training time per epoch of SGFormer when the layer number of global attention increases from one to more. We found that using more layers does

not contribute to considerable performance boost and instead leads to performance drop on the heterophilic graph `actor`. Even worse, multi-layer attention requires more training time costs. Notably, using one-layer attention of SGFormer can consistently yield highly competitive performance as the multi-layer attention. These results verify the effectiveness of our one-layer attention that has desirable expressivity and superior efficiency. In Sec. 6 we will shed more insights into why the one-layer all-pair attention is an expressive model.

**How does one-layer global attention of other models perform compared to the multi-layer counterparts?** Fig. 4 presents the performance of NodeFormer, SGFormer w/o self-loop (removing the self-loop propagation in Eqn. 5) and SGFormer w/ Softmax (replacing our attention by the Softmax attention), w.r.t. different numbers of attention layers in respective models. We found that using one-layer attention for these models can yield decent results in quite a few cases, which suggests that for other implementations of global attention, using a single-layer model also has potential for competitive performance. In some cases, for instance, NodeFormer produces unsatisfactory results on `ogbn-proteins` with one layer. This is possibly because NodeFormer couples the global attention and local propagation in each layer, and using the one-layer model could sacrifice the efficacy of the latter. On top of all of these, we can see that it can be a promising future direction for exploring effective shallow attention models that can work consistently and stably well. We also found when using deeper model depth, SGFormer w/o self-loop exhibits clear performance degradation and much worse than the results in Fig. 3, which suggests that the self-loop propagation in Eqn. 5 can help to maintain the competitiveness of SGFormer with multi-layer attentions.

## 6 Theoretical Justifications

To resolve lingering opaqueness on the rationale of our methodology, we attempt to shed some insights on why the one-layer attention model can be expressive enough for learning desirable representations from global interactions. We mainly consider the feature propagation step of one Transformer layer:

$$\mathbf{z}_u^{(k)} = (1-\tau)\mathbf{z}_u^{(k-1)} + \tau \sum_{v=1}^{N} c_{uv}^{(k)} \mathbf{z}_v^{(k-1)}, \qquad (5)$$

where $c_{uv}^{(k)}$ is the attention score between node $u$ and $v$ at the $k$-th layer and $0 < \tau \leq 1$ is a weight. For example, in the original Transformers [50], $c_{uv}^{(k)}$ is the Softmax-based attention score. In our model Eqn. 3, we correspondingly have[2]: $c_{uu}^{(k)} = \frac{N+\tilde{\mathbf{q}}_u^\top \tilde{\mathbf{k}}_u}{N+\sum_w \tilde{\mathbf{q}}_u^\top \tilde{\mathbf{k}}_w}$ and $c_{uv}^{(k)} = \frac{\tilde{\mathbf{q}}_u^\top \tilde{\mathbf{k}}_v}{N+\sum_w \tilde{\mathbf{q}}_u^\top \tilde{\mathbf{k}}_w}$ for $v \neq u$. Our analysis is mainly based on the interpretation of one Transformer layer as an optimization step for a principled graph signal denoising problem [47; 21; 13] with a particular objective. The denoising problem defined over $N$ nodes in a system aims at smoothing the node features by adaptively absorbing the information from other nodes. The denoised features synthesize the initial information and the global information under the guidance of the given objective.

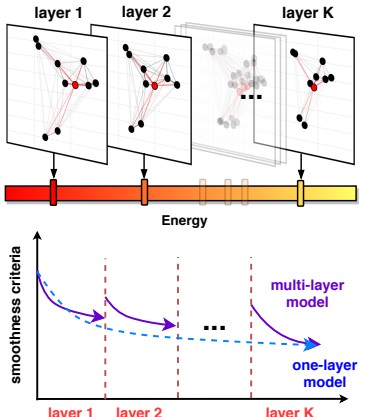

Figure 5: Illustration of the theoretical analysis showing the equivalence between the multi-layer and one-layer attention models. The one-layer attention model can produce the same effect on the smoothness criteria and help to save potential redundancy.

**Theorem 1.** *For any given attention matrix $\mathbf{C}^{(k)} = [c_{uv}^{(k)}]_{N \times N}$, Eqn. 5 is equivalent to a gradient descent operation with step size $\frac{\tau}{2\lambda}$ for an optimization problem with the cost function:*

$$\min_{\mathbf{Z}} \sum_u \|\mathbf{z}_u - \mathbf{z}_u^{(k-1)}\|_2^2 + \lambda \sum_{u,v} c_{uv}^{(k)} \|\mathbf{z}_u - \mathbf{z}_v\|_2^2, \qquad (6)$$

*where $\lambda$ is a trading weight parameter for the local smoothness and global smoothness criteria [69].*

---

[2]For analysis and comparing the one-layer and multi-layer attention models, we extend our model to have arbitrary $K$ layers, where each layer comprises an attention network of Eqn. 3 with independent parameterization, and use the superscript $k$ to index each layer.

Eqn. 6 can be seen as a generalization of the Dirichlet energy [70] defined over a dense attention graph, which is also utilized as the optimization target in the energy-constrained graph diffusion framework for designing principled and scalable attention layers [54]. Specifically, the pair-wise attention score $c_{uv}^{(k)}$ reflects the proximity between nodes $u$ and $v$, and the attention matrix $\mathbf{C}^{(k)}$ enforces adaptive denoising effect at the $k$-th layer. Since $\mathbf{C}^{(k)}$ varies layer by layer, multi-layer attention models can be seen as a cascade of descent steps on layer-dependent denoising objectives. As we will show in the next result, the multi-layer model can be simplified as a single-layer one, while the latter contributes to the same denoising effect.

**Theorem 2.** *For any $K$-layer attention model (where $K$ is an arbitrary positive integer) with the layer-wise updating rule defined by Eqn. 5, there exists $\mathbf{C}^* = [c_{uv}^*]_{N \times N}$ such that one gradient descent step for the optimization problem (from the initial embeddings $\mathbf{Z}^{(0)} = [\mathbf{z}_u^{(0)}]_{u=1}^N$)*

$$\min_{\mathbf{Z}} \sum_u \|\mathbf{z}_u - \mathbf{z}_u^{(0)}\|_2^2 + \lambda \sum_{u,v} c_{uv}^* \|\mathbf{z}_u - \mathbf{z}_v\|_2^2, \tag{7}$$

*can yield the output embeddings $\mathbf{Z}^{(K)} = [\mathbf{z}_u^{(K)}]_{u=1}^N$ of the $K$-layer model.*

The above result indicates that for any multi-layer attention model, one can always find a single-layer attention model that behaves in the same way aggregating the global information. Moreover, the multi-layer model optimizes different objectives at each layer, which suggests potential redundancy compared to the one-layer model that pursues a fixed objective in one step. And, according to Theorem 1, the one-layer model contributes to a steepest decent step on this objective. We present an illustration of our theoretical results in Fig. 5.

## 7    Conclusions and Outlooks

This paper explores the potential of simple Transformer-style architectures for learning large-graph representations where the scalability challenge plays a bottleneck. Through extensive experiments across diverse node property prediction benchmarks whose graph sizes range from thousands to billions, we demonstrate that a one-layer attention model combined with a vanilla GCN can surprisingly produce highly competitive performance. The simple and lightweight architecture enables our model, dubbed as SGFormer, to scale smoothly to an extremely large graph with 0.1B nodes and yields 30x acceleration compared to state-of-the-art Transformers on medium-sized graphs. We also provide accompanying theoretical justification for our methodology. On top of our technical contributions, we believe the results in this paper could shed lights on a new promising direction for building powerful and scalable Transformers on large graphs, which is largely under-explored.

**Future Works.** One limitation of the present work lies in the tacit assumption that the training and testing data comes from the identical distribution. Though this assumption is commonly adopted for research on representation learning where the main focus is the expressiveness and representation power of the models, in practical scenarios, testing data may come from previously unseen distributions, resulting in the challenge of out-of-distribution learning. Promising future directions could be to explore the potential of graph Transformers in out-of-distribution generalization tasks where distribution shifts exist [56; 62], and the capabilities of graph Transformers for detecting out-of-distribution samples [53; 30]. We leave more studies along these under-explored, orthogonal areas as future works.

Furthermore, in this work, we follow the common practice and mainly focus on the benchmark settings for evaluating the model w.r.t. a fundamental challenge of learning representations on large graphs. While in principle Transformers can be applied for other graph-based tasks and more broadly domain-specific applications, such as recommender systems [55], circuit designs [6] and combinatorial optimization [28; 29], these scenarios often require extra task-dependent designs. We thereby leave exploration along this orthogonal direction aiming to unlock the potential of graph Transformers for various specific applications as future works.

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

# A  Proof for Technical Results

## A.1  Proof for Theorem 1

**Theorem 1.** *For a given attention matrix $\mathbf{C}^{(k)} = [c_{uv}^{(k)}]_{N \times N}$, Eqn. 5 is equivalent to a gradient descent with step size $\frac{\tau}{2\lambda}$ for an optimization problem with the cost function:*

$$\min_{\mathbf{Z}} \sum_u \|\mathbf{z}_u - \mathbf{z}_u^{(k-1)}\|_2^2 + \lambda \sum_{u,v} c_{uv}^{(k)} \|\mathbf{z}_u - \mathbf{z}_v\|_2^2, \tag{8}$$

*where $\lambda$ is a trading weight for the local smoothness and global smoothness criteria [69].*

*Proof.* We denote the cost function at the $k$-th layer as

$$E(\mathbf{Z}; \mathbf{Z}^{(k-1)}) = \sum_u \|\mathbf{z}_u - \mathbf{z}_u^{(k-1)}\|_2^2 + \lambda \sum_{u,v} c_{uv}^{(k)} \|\mathbf{z}_u - \mathbf{z}_v\|_2^2. \tag{9}$$

The gradient of $E(\mathbf{Z}; \mathbf{Z}^{(k-1)})$ w.r.t. $\mathbf{z}_u$ can be computed by

$$\frac{\partial E(\mathbf{Z}; \mathbf{Z}^{(k-1)})}{\partial \mathbf{z}_u} = 2(\mathbf{z}_u - \mathbf{z}_u^{(k-1)}) + 2\lambda \sum_{u,v} c_{uv}^{(k)} (\mathbf{z}_u - \mathbf{z}_v). \tag{10}$$

The gradient descent with step size $\frac{\tau}{2\lambda}$ that minimizes the cost function $E(\mathbf{Z}; \mathbf{Z}^{(k-1)})$ at the current layer is

$$\begin{aligned}
\mathbf{z}_u^{(k)} &= \mathbf{z}_u^{(k-1)} - \tau \left. \frac{\partial E(\mathbf{Z}; \mathbf{Z}^{(k-1)})}{\partial \mathbf{z}_u} \right|_{\mathbf{Z} = \mathbf{Z}^{(k-1)}} \\
&= \mathbf{z}_u^{(k-1)} - 2\frac{\tau}{2\lambda}(\mathbf{z}_u^{(k-1)} - \mathbf{z}_u^{(k-1)}) - 2\frac{\tau}{2\lambda}\lambda \sum_v c_{uv}^{(k)} (\mathbf{z}_u^{(k-1)} - \mathbf{z}_v^{(k-1)}) \\
&= (1 - \tau) \sum_v c_{uv}^{(k)} \mathbf{z}_u^{(k-1)} + \tau \sum_v c_{uv}^{(k)} \mathbf{z}_v^{(k-1)} \\
&= (1 - \tau) \mathbf{z}_u^{(k-1)} + \tau \sum_v c_{uv}^{(k)} \mathbf{z}_v^{(k-1)}.
\end{aligned} \tag{11}$$

The last step is due to the normalization of the attention scores, i.e., $\sum_v c_{uv}^{(k)} = 1$. Eqn. 18 is the updating equation of the $k$-th layer of Transformers, i.e., Eqn. 5. $\qquad\square$

## A.2  Proof for Theorem 2

**Theorem 2.** *For any $K$-layer attention model (where $K$ is an arbitrary positive integer) with the layer-wise updating rule defined by Eqn. 5, there exists $\mathbf{C}^* = [c_{uv}^*]_{N \times N}$ such that one gradient descent step for the optimization problem (from the initial embeddings $\mathbf{Z}^{(0)} = [\mathbf{z}_u^{(0)}]_{u=1}^N$)*

$$\min_{\mathbf{Z}} \sum_u \|\mathbf{z}_u - \mathbf{z}_u^{(0)}\|_2^2 + \lambda \sum_{u,v} c_{uv}^* \|\mathbf{z}_u - \mathbf{z}_v\|_2^2, \tag{12}$$

*can yield the output embeddings $\mathbf{Z}^{(K)} = [\mathbf{z}_u^{(K)}]_{u=1}^N$ of the $K$-layer model.*

*Proof.* Assume the $K$-layer model with the layer-wise updating rule defined by Eqn. 5 yields a sequence of attention matrices and node embeddings $\mathbf{C}^{(1)}, \mathbf{Z}^{(1)}, \mathbf{C}^{(2)}, \mathbf{Z}^{(2)}, \cdots, \mathbf{C}^{(K)}, \mathbf{Z}^{(K)}$. We define a propagation matrix $\mathbf{P}^{(k)}$:

$$\mathbf{P}^{(k)} = (1 - \alpha) \begin{bmatrix} 1 & 0 & \cdots & 0 \\ 0 & 1 & \cdots & 0 \\ \vdots & \vdots & \ddots & \vdots \\ 0 & 0 & \cdots & 1 \end{bmatrix} + \alpha \begin{bmatrix} c_{11}^{(k)} & c_{12}^{(k)} & \cdots & c_{1N}^{(k)} \\ c_{21}^{(k)} & c_{22}^{(k)} & \cdots & c_{2N}^{(k)} \\ \vdots & \vdots & \ddots & \vdots \\ c_{N1}^{(k)} & c_{N2}^{(k)} & \cdots & c_{NN}^{(k)} \end{bmatrix} = (1 - \alpha)\mathbf{I} + \alpha \mathbf{C}^{(k)}. \tag{13}$$

Then Eqn. 5 can be equivalently written as

$$\mathbf{Z}^{(k)} = \mathbf{P}^{(k)}\mathbf{Z}^{(k-1)}. \tag{14}$$

By stacking $K$ layers of propagation, we can denote the output embeddings as

$$\mathbf{Z}^{(K)} = \mathbf{P}^{(k)}\mathbf{Z}^{(K-1)} = \mathbf{P}^{(K)}\mathbf{P}^{(k-1)}\mathbf{Z}^{(k-2)} = \cdots = \mathbf{P}^{(K)}\cdots\mathbf{P}^{(1)}\mathbf{Z}^{(0)} = \mathbf{P}^{*}\mathbf{Z}^{(0)}. \tag{15}$$

We next conclude the proof by construction. Defining $\mathbf{C}^{*} = \frac{1}{\tau^{*}}\left(\mathbf{P}^{*} - (1-\tau^{*})\mathbf{I}\right)$, where in specific,

$$\mathbf{C}^{*} = \begin{bmatrix} c_{11}^{*} & c_{12}^{*} & \cdots & c_{1N}^{*} \\ c_{21}^{*} & c_{22}^{*} & \cdots & c_{2N}^{*} \\ \vdots & \vdots & \ddots & \vdots \\ c_{N1}^{*} & c_{N2}^{*} & \cdots & c_{NN}^{*} \end{bmatrix} = \frac{1}{\tau^{*}}\left(\begin{bmatrix} p_{11}^{*} & p_{12}^{*} & \cdots & p_{1N}^{*} \\ p_{21}^{*} & p_{22}^{*} & \cdots & p_{2N}^{*} \\ \vdots & \vdots & \ddots & \vdots \\ p_{N1}^{*} & p_{N2}^{*} & \cdots & p_{NN}^{*} \end{bmatrix} - (1-\tau^{*})\begin{bmatrix} 1 & 0 & \cdots & 0 \\ 0 & 1 & \cdots & 0 \\ \vdots & \vdots & \ddots & \vdots \\ 0 & 0 & \cdots & 1 \end{bmatrix}\right). \tag{16}$$

We can show that solving the denoising problem with gradient step size $\frac{\tau^{*}}{2\lambda}$ w.r.t. the objective

$$\min_{\mathbf{Z}} \sum_{u} \|\mathbf{z}_u - \mathbf{z}_u^{(0)}\|_2^2 + \lambda \sum_{u,v} c_{uv}^{*} \|\mathbf{z}_u - \mathbf{z}_v\|_2^2, \tag{17}$$

will induce the output embeddings $\mathbf{Z}^{(K)}$, by noticing that

$$
\begin{aligned}
&\mathbf{z}_u^{(0)} - \frac{\tau^{*}}{2\lambda}\left.\frac{\partial E(\mathbf{Z};\mathbf{Z}^{(0)})}{\partial \mathbf{z}_u}\right|_{\mathbf{Z}=\mathbf{Z}^{(k-1)}} \\
=&\mathbf{z}_u^{(0)} - 2\frac{\tau^{*}}{2\lambda}(\mathbf{z}_u^{(0)} - \mathbf{z}_u^{(0)}) - 2\frac{\tau^{*}}{2\lambda}\lambda\sum_{v} c_{uv}^{*}(\mathbf{z}_u^{(0)} - \mathbf{z}_v^{(0)}) \\
=&(1-\tau^{*})\sum_{v} c_{uv}^{*}\mathbf{z}_u^{(0)} + \tau^{*}\sum_{v} c_{uv}^{*}\mathbf{z}_v^{(0)} \\
=&(1-\tau^{*})\mathbf{z}_u^{(0)} + \tau\sum_{v} c_{uv}^{*}\mathbf{z}_v^{(0)} \\
=&(1-\tau^{*})\mathbf{z}_u^{(0)} + \tau^{*}\left(\frac{1}{\tau^{*}}p_{uu}^{*} + 1 - \frac{1}{\tau^{*}}\right)\mathbf{z}_u^{(0)} + \tau^{*}\sum_{v\neq u}\frac{1}{\tau^{*}}p_{uv}^{*}\mathbf{z}_v^{(0)} \\
=&\sum_{v} p_{uv}^{*}\mathbf{z}_v^{(0)}.
\end{aligned} \tag{18}
$$

And, we have $\sum_{v} p_{uv}^{*}\mathbf{z}_v^{(0)} = \mathbf{z}_u^{(K)}$ since by definition $\mathbf{Z}^{(K)} = \mathbf{P}^{*}\mathbf{Z}^{(0)}$. $\qquad\square$

# B Dataset Information

Table 5: Information for node property prediction datasets.

| Dataset | Property | # Tasks | # Nodes | # Edges | # Feats | # Classes |
|---|---|---|---|---|---|---|
| Cora | homophilous | 1 | 2,708 | 5,429 | 1,433 | 7 |
| CiteSeer | homophilous | 1 | 3,327 | 4,732 | 3,703 | 6 |
| PubMed | homophilous | 1 | 19,717 | 44,324 | 500 | 3 |
| Actor | heterophilic | 1 | 7,600 | 29,926 | 931 | 5 |
| Squirrel | heterophilic | 1 | 5,201 | 216,933 | 2,089 | 5 |
| Chameleon | heterophilic | 1 | 2,277 | 36,101 | 2,325 | 5 |
| Deezer | heterophilic | 1 | 28,281 | 92,752 | 31,241 | 2 |
| ogbn-proteins | multi-task | 112 | 132,534 | 39,561,252 | 8 | 2 |
| Amazon2M | long-range dependence | 1 | 2,449,029 | 61,859,140 | 100 | 47 |
| pokec | heterophilic | 1 | 1,632,803 | 30,622,564 | 65 | 2 |
| ogbn-arxiv | homophilous | 1 | 169,343 | 1,166,243 | 128 | 40 |
| ogbn-papers100M | homophilous | 1 | 111,059,956 | 1,615,685,872 | 128 | 172 |

Our experiments are based on 12 real-world datasets which are all publicly available with open access. The information of these datasets is presented in Table 5.

- `cora`, `citeseer` and `pubmed` [46] are three networks that contain citations, with nodes representing documents and edges representing citation links. The features of the nodes are represented as bag-of-words, capturing the content of the documents. The goal is to predict the academic topic of each paper. We follow the semi-supervised settings in [23], using randomly sampled 20 instances per class as training set, 500 instances as validation set, and 1,000 instances as testing set.

- `actor` is the actor-only induced subgraph of the film-directoractor-writer network [49]. In this dataset, each node corresponds to an actor, and the edges between nodes signify their co-occurrence on the same Wikipedia page. The node features are comprised of keywords extracted from the respective actors' Wikipedia pages. The goal is to classify the nodes into five categories based on the content of the actors' Wikipedia pages. This dataset exhibits a relatively low homophily ratio and belongs to a heterophilic graph.

- `squirrel` and `chameleon` are two page-page networks that focus on on specific topics in Wikipedia [41]. In these datasets, the nodes represent web pages, and the edges denote mutual links between the pages. The node features are composed of a selection of informative nouns extracted from the corresponding Wikipedia pages. The goal is to categorize the nodes into five distinct groups based on the average monthly traffic received by each web page. Furthermore, these datasets are recognized as heterophilic graphs, indicating the presence of connections between nodes with distinct labels. A recent paper [38] indicates that the original split of these datasets introduces overlapping nodes between training and testing, and propose a new data split that filters out the overlapping nodes. We use its provided split for evaluation.

- `deezer-europe` is a user–user network of European members of the music streaming service Deezer [42]. In this network, the nodes represent users, while the edges signify mutual friendships between these users. The node features are based on the artists that the streamers have liked. The goal is to classify the gender of the users. We follow the benchmark setting in [32] and use a random 50%/25%/25% train/valid/test split.

- `ogbn-proteins` consists of a graph where nodes correspond to proteins and edges indicate various biologically significant associations between proteins, categorized by their types (based on species). Each edge is accompanied by 8-dimensional features, with each dimension representing the approximate confidence level of a specific association type and ranging from 0 to 1. The proteins in the dataset originate from 8 different species. The goal is to predict the presence or absence of 112 protein functions using a multi-label binary classification approach. We use the public OGB split [19].

- `ogbn-arxiv` is a citation network among all Computer Science (CS) Arxiv papers, as described by [19]. In this network, each node corresponds to an Arxiv paper, and the edges indicate the citations between papers. Each paper is associated with a 128-dimensional feature vector, obtained by averaging the word embeddings of its title and abstract. The word embeddings are generated using the WORD2VEC model. The objective is to predict the subject areas of Arxiv CS papers, specifically targeting 40 distinct subject areas. We adopt the split of [19], which involves training the model on papers published until 2017, validating on those published in 2018, and testing on papers published from 2019 onwards.

- `Amazon2M` is derived from the Amazon Co-Purchasing network [33]. In this dataset, each node represents a product, and the presence of a graph link between two products indicates that they are frequently purchased together. The node features are generated by extracting bag-of-word features from the product descriptions. The labels assigned to the products are based on the top-level categories they belong to. We follow the recent work [57] using random split with the ratio 50%/25%/25%.

- `pokec` [26] is a large-scale social network dataset that encompasses various features, including profile information like geographical region, registration time, and age. The goal is to predict the gender of users based on these features. For data split, we randomly divide the nodes into training, validation, and testing sets, with ratios of 10%, 10%, and 80%, respectively.

- `ogbn-papers100M` is a gigantic web-scale graph that consists of 111 million papers as nodes. The difference of this dataset from others is that not all nodes are labeled and the labeled ratio is relatively low. The labeled portion comprises approximately 1.5 million

ARXIV papers, each of which is labeled with one subject area. The goal is to classify the paper into 172 ARXIV subject areas. We also follow the public split by OGB [19]. This dataset is nearly the largest public graph benchmark.

## C   Implementation Details

In this section, we provide more details regarding the model implementation, including model architectures, hyper-parameter settings and training details, to facilitate the reproducibility.

### C.1   Model Architectures

Our model is comprised of four modules: input layer, global attention, GNN model, and output layer. We present the details for each of them as follows.

- The input layer is a one-layer MLP with non-linear activation that transforms the input features $\mathbf{X} \in \mathbb{R}^{N \times D}$ to the node embeddings in the latent space $\mathbf{Z}^{(0)} \in \mathbb{R}^{N \times d}$.

- The global attention computes the all-pair influence among $N$ nodes through the attention mechanism as described in Eqn. 3. Specifically, it feeds the initial embeddings $\mathbf{Z}^{(0)} \in \mathbb{R}^{N \times d}$ into an one-layer attention network and outputs the updated embeddings $\mathbf{Z} \in \mathbb{R}^{N \times d}$ that absorbs the global information from other nodes.

- The GNN network GN is instantiated as a graph convolution network which takes the input graph $\mathbf{A}$ and the initial embeddings $\mathbf{Z}^{(0)} \in \mathbb{R}^{N \times d}$ as input and outputs the updated embeddings for each node. The latter is then linearly added with the output embeddings of the global attention, i.e., $\mathbf{Z}_O = (1 - \alpha)\mathbf{Z} + \alpha \text{GN}(\mathbf{Z}^{(0)}, \mathbf{A})$, as the final representation for each node.

- The output layer is a feed-forward layer for prediction, which maps the node representations $\mathbf{Z}_O \in \mathbb{R}^{N \times d}$ into the predicted labels $\hat{Y}$. The prediction $\hat{Y}$ will be used for computing the standard loss function of the form $\sum_u l(\hat{y}_u, y_u)$, where $l(\cdot, \cdot)$ can be cross-entropy for classification tasks or mean square error for regression tasks.

### C.2   Training Details

We use different training schemes for graph datasets of different scales. For relatively small graphs (e.g., from 1K to 0.1M nodes), we use full-graph training; for larger graphs that cannot be trained in a single GPU, we consider mini-batch training. The training details are introduced below.

**Full-Graph Training.** For all the datasets presented in Table 2 and `ogbn-arxiv`, we use the full-graph training. In specific, we feed the whole graph into the model and compute the all-pair attention among all the nodes and predict their labels for loss computation. For inference, similarly, we use the whole graph as input and compute the metric based on the prediction for the validation/testing nodes.

**Mini-Batch Training.** For datasets (except `ogbn-arxiv`) presented in Table 3, due to the large graph sizes, we consider the mini-batch training scheme used in [57]. Specifically, at the beginning of each training epoch, we randomly shuffle the nodes and partition the nodes into mini-batches with the size $B$. Then in each iteration, we feed one mini-batch (the input graph among these $B$ nodes are directly extracted by the subgraph of the original graph) into the model for loss computation on the training nodes within this mini-batch. Then for inference on `ogbn-proteins`, `Amazon2M` and `pokec` (with 0.1M to 1M nodes), following the pipeline in [19] for evaluation, we can feed the whole graph into the model using CPU, which computes the all-pair attention among all the nodes in the dataset. For the gigantic graph `ogbn-papers100M` that cannot be fed as whole into the CPU with moderate memory during inference, we adopt the mini-batch partition strategy used in training to reduce the overhead. Since our model allows large batch sizes, we found the mini-batch partition can yield decent performance. In specific, we set the batch size as 10K, 0.1M, 0.1M and 0.4M for `ogbn-proteins`, `Amazon2M`, `pokec` and `ogbn-papers100M`, respectively.

**Evaluation Protocol.** We follow the common practice for node classification tasks and set a fixed number of training epochs: 300 for medium-sized graphs, 1000 for large-sized graphs, and 50 for the extremely large graph `ogbn-papers100M`. The testing accuracy achieved by the model that reports

the highest result on the validation set is used for evaluation. We run each experiment with five independent trials using different initializations, and report the mean and variance of the metrics.

## C.3 Hyper-parameters

We use the model performance on the validation set for hyper-parameter settings of all models including the competitors. Unless otherwise stated, the hyper-parameters are selected using grid search with the searching space:

- learning rate within $\{0.001, 0.005, 0.01, 0.05, 0.1\}$;
- weight decay within $\{1e-5, 1e-4, 5e-4, 1e-3, 1e-2\}$;
- hidden size within $\{32, 64, 128, 256\}$;
- dropout ratio within $\{0, 0.2, 0.3, 0.5\}$;
- number of layers within $\{1, 2, 3\}$.

The searching space for other hyper-parameters that differ in each model is as follows:

- GAT: head number $\in \{4, 8, 16\}$.
- SGC: hop number $\in \{1, 2, 3\}$.
- APPNP: $\alpha \in \{0.1, 0.2, 0.5, 0.9\}$.
- JKNet: jumping knowledge type $\in \{\text{max, concatenation}\}$.
- CPGNN: pretraining epochs $\in \{100, 200, 300\}$.
- SIGN: hop number $\in \{1, 2, 3\}$.
- GloGNN: $\alpha \in [0, 1]$, $\beta_1 \in \{0, 1, 10\}$, $\beta_2 \in \{01, 1, 10, 100, 1000\}$, $\gamma \in [0, 0.9]$, norm_layers $\in \{1, 2, 3\}$, $K \in [1, 6]$.
- Graphormer: embedding dimension $\{64, 128, 256, 512\}$.
- GraphTrans: The dimension of the attentive module $\{64, 128, 256\}$, the dimension of the GCN $\{16, 32, 64, 128\}$.
- NodeFormer: rb_order $\in \{1, 2, 3\}$, hidden channels $\in \{16, 32, 64, 128\}$, edge regularization weight $\in [0, 1]$, head number $\in \{1, 2, 3, 4\}$.
- SGFormer: number of global attention layers is fixed as 1, number of GCN layers $\in \{1, 2, 3\}$, weight $\alpha$ within $\{0.5, 0.8\}$.

# D  More Empirical Results

In this section, we supplement more experimental results to further study what patterns the global attention learn from data. We visualize the attention matrices given by SGFormer on different datasets, and the results are shown in Fig. 6. We found that overall, the attention graphs are quite different from the input graphs, which implies that the global attention mechanism indeed captures certain informative patterns from data that are helpful for downstream prediction. Furthermore, since our model uses one-layer attention, the attention scores estimated by the model can be used for interpreting the influence among samples. For instance, as shown by the results, there are some sharp lines in the heat map which corresponds to the nodes that are recognized as important 'information source' by the model and allocated with large attention weights. These nodes are akin to some hubs in the network that exert much influence on other nodes.

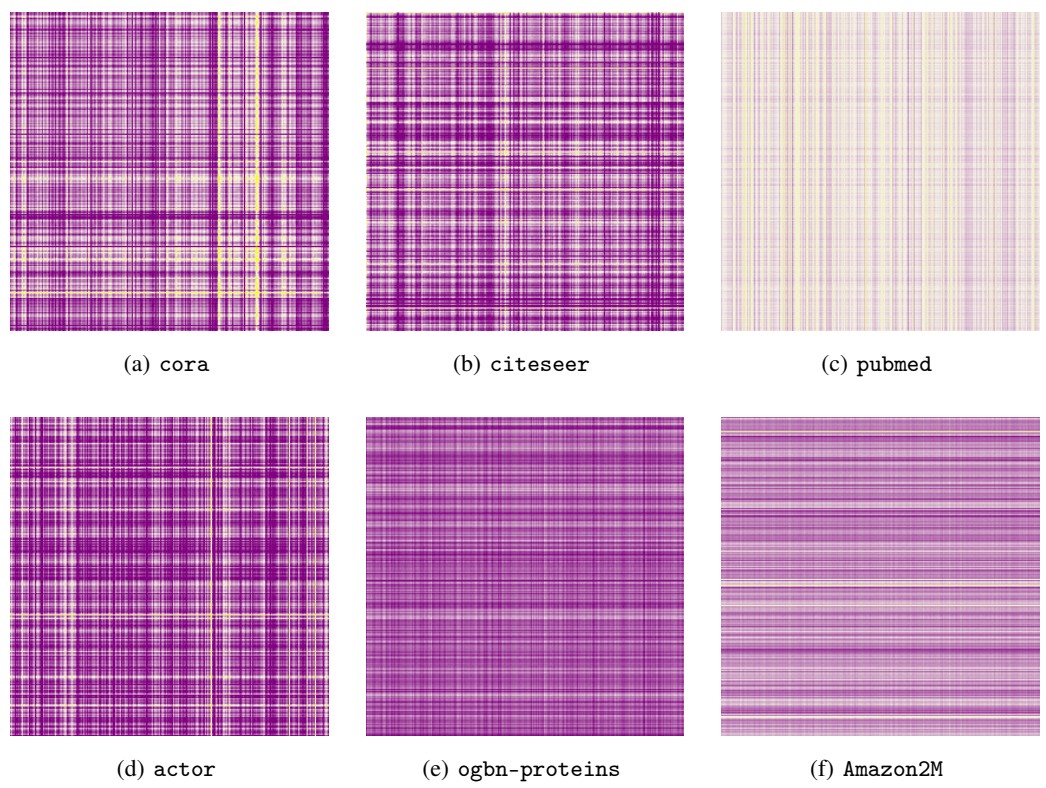

(a) cora       (b) citeseer       (c) pubmed

(d) actor       (e) ogbn-proteins       (f) Amazon2M

Figure 6: Visualization of the attention weights produced by SGFormer.

