# OpenReview forum: "SGFormer: Simplifying and Empowering Transformers for Large-Graph Representations"
_NeurIPS.cc/2023/Conference — NeurIPS 2023 poster_

### Official Review · Reviewer_bWXq · 2023-06-27

**Soundness:** 2 fair
**Presentation:** 2 fair
**Contribution:** 1 poor
**Rating:** 5
**Confidence:** 4

**Summary:**

Paper proposes a linear attention mechanism in the transformers for graph data. The attention becomes linear by eliminating the softmax in the dot product attention and multiplying K (transposed) and V matrices first followed by multiplication with Q matrix. The Q, K matrices are normalized by their respective frobenius norms to prevent explosion in the dot products. Authors further propose using a single attention layer instead of multiple layers. This they claim is sufficient over multiple layers from a signal denoising perspective. Experimental results demonstrate comparable results with baselines. The linear attention mechanism helps in reducing training compared to baselines.

**Strengths:**

1) The model is shown to be efficient compared to the standard scaled dot product attention which enables scaling to graphs with millions of nodes.
2) The results are comparable with baselines

**Weaknesses:**

1) Considering that the essence of the method lies in the linear attention which is obtained by eliminating nonlinearity and using the associative property of matrix multiplication, the novelty is limited. Further analysis needs to be done on what is the benefit/deterioration due to the elimination of softmax in isolation.
2) The gains for the small datasets are marginal, in some cases within deviation. In the large datasets too it appears that the optimal baseline results have been omitted. For example in the ogbn-papers dataset there are efficient transformers (https://arxiv.org/abs/2009.03509) that achieve better results than reported in this paper. Also on the same dataset SIGN method reports best result of 69.84 but lesser value is reported. Since the proposed method is not a pure transformer but a mix of GNN and transformer I would also suggest trying GraphGPS on the same GNN and another efficient transformer on these large datasets as a fair baseline.
3) While the method claims to not be using PEs, the authors induce representations learnt from a GNN which indirectly acts as a PE. This is similar to the framework of GraphGPS and the claim doesn’t seem to hold.
4) In continuation to the above point, the results without GNN are very poor and so it seems that it is not the transformer alone that gives the gain but the combination with the GNN. Thus the results may not just be from the single layer linear attention as claimed and the claim should be modified in accordance with the experimental evidence.
5) The model with linear attention also seems to be having problems scaling to large graphs and the authors resort to sampling techniques from Nodeformer for the same. Since efficiency is the main proposed contribution of the paper it is expected in principle to be integrated in the method rather than induced externally.

Missing Citations:
1) https://proceedings.mlr.press/v162/choromanski22a.html

**Questions:**

1) In the large datasets it appears that the optimal baseline results have been omitted. For example in the ogbn-papers dataset there are efficient transformers (https://arxiv.org/abs/2009.03509) that achieve better results than reported in this paper. Also on the same dataset SIGN method reports best result of 69.84 but lesser value is reported. Since the proposed method is not a pure transformer but a mix of GNN and transformer have the authors tried inducing the proposed linear attention in GraphGPS framework for the large datasets?
2) While the method claims to not be using PEs, the authors induce representations learnt from a GNN which indirectly acts as a PE. This is similar to the framework of GraphGPS and the claim doesn’t seem to hold. I would expect further clarification on why the GNN is not a PE and whether the proposed method can work independent of the GNN (which does not seem to be the case from the ablations in appendix).
3) While it is understandable that eliminating softmax helps gaining efficiency, are there any drawbacks? For example is the attention now more dense or some attention matrix values cannot be achieved or are there some other issues? The authors show an ablation that results with softmax are poor but this is in the entire architecture including the GNN etc. I would suggest performing an isolated study on the importance of and impact of eliminating softmax.
4) Considering the method uses a single layer of attention, is it expressive enough to learn complex functions and permit pretraining? For example if the authors would pretrain the existing method on a large scale dataset and finetune on a  smaller dataset would it generalize as graphormer and related methods have shown?
5) In theorem 1 the authors mention that equation 5 is the gradient descent solution to the low pass smoothing objective. But what if the objective is to learn some other function that does not have a low frequency characteristic? Would multiple attention heads still not help in that case?

**Limitations:**

The limitations of the method have been addressed in the paper.

---

> ### Author Rebuttal · Authors · 2023-08-08
>
> Thank you for the valuable feedback. Below we group related points in Weaknesses (W) and Questions (Q).
>
> > **Q1 & W2: Optimal baseline results**
>
> We have to respectfully point out the factual errors from the reviews. **The suggested work [1] does not experiment on the largest graph ogbn-papers100M**, and for ogbn-proteins, [1] uses their own splits (by subgraph partition) instead of standard OGB splits, so **not directly comparable to us**. Besides, **SIGN [2] reports the optimal test acc 65.11±0.14 on ogbn-papers100M (see Table 8 of [2]), rather than 69.84 as suggested by the reviewer (which should be SIGN's validation acc)**. In Table 3, our test score 66.01 ± 0.37 is significantly higher than their optimal test acc. Given the benchmark setting we used and fair comparison with strong competitors (including two recent efficient graph Transformers NodeFormer/ANS-GT), our achieved gains are non-trivial, as resonated by other reviewers.
>
> [1] Masked Label Prediction: Unified Message Passing Model for Semi-Supervised Classification
>
> [2] SIGN: Scalable Inception Graph Neural Networks
>
> > **Q1 & W2: Compare with GraphGPS and other attentions**
>
> We tried replacing our attention with GraphGPS's on large graphs in Table 3, and unfortunately, the model suffers OOM with a 24GB GPU. The scalability issue of GraphGPS on these large-sized graphs is also encountered by [3] (see their Table 2). Nevertheless, we add comparison with GraphGPS on medium-sized graphs, and found our model achieves significantly higher scores (see results in Table 1 of the uploaded PDF).
>
> For other attentions (NodeFormer and GAT), we have compared with them in Table 6 which again shows the superiority of our attention.
>
> [3] GOAT: A Global Transformer on Large-scale Graphs
>
> > **Q2 & W3: Positional embeddings (PE) and role of GNNs**
>
> Our claim "the model does not require PE" is not violated by the used GNNs, since **GNNs are commonly not recognized as PE in existing literature**. E.g. GraphTrans [4] uses GNNs as a module and claims their model does not require PE. In common sense, PE often refers to the Laplacian features for each node (absolute PE [5]) or distance encodings for node pairs (relative PE [6]), different from GNNs. The PE and GNNs are also distinguished by [7].
>
> [4] Representing long-range context for graph neural networks with global attention
>
> [5] A generalization of transformer networks to graphs
>
> [6] Do transformers really perform bad for graph representation?
>
> [7] Transformer for graphs: An overview from architecture perspective
>
> > **Q2 & W4: Performance degrades without GNNs**
>
> It is natural to see performance drop w/o GNN, since in our model GNN can additionally use input graphs complementary to global attention. Having said that **the one-layer global attention plays a crucial role in our model** for achieving its first-rank results in Table 2 and 3 over pure GNNs and powerful Transformers (including GraphTrans that also uses GNN). This suggests our proposed attention contributes to the gains that are much more difficult to achieve over the GNN baseline.
>
> Furthermore, **our claim "a one-layer attention can bring up competitive performance" is originally stated within the context of graph Transformers, wherein peer models use deep attentions and PE (or GNN as an alternative). Our viewpoint lies in the advances over existing models, and our simple attention can significantly enhance the scalability on large graphs**. Based on our empirical evidence and acknowledgement by other reviewers, we believe this claim is properly justified.
>
> > **Q3 & W1: Compare with softmax attention**
>
> The softmax attention that requires $O(N^2)$ cannot scale to large graphs, so we add some comparison on small graphs in Table 2 of the uploaded PDF. It shows softmax attention yields undesired results, possibly because the softmax would lead to gradient vanishing when N is large, as observed by NodeFormer. In terms of our attention, we do not observe any obvious inability (see some visualization results in Fig. 1 of the uploaded PDF).
>
> > **Q4: Expressiveness of one-layer attention**
>
> The one-layer attention is expressive enough for learning arbitrary all-pair interactions as done by the multi-layer model, as discussed in Sec 6 from signal denoisng perspective. Also notice that in our model, the one-layer attention does not mean using one-layer neural network, and one can stack multiple feedforward NN layers on top of the attention if needed. In this way, **the model can still have enough capacity for learning complex functions**.
>
> For pretraining tasks, which is orthogonal to our current focus as well as the peer models we compare in this area, we leave this extension as future works.
>
> > **Q5: Low-pass filter for Eqn. 5**
>
> Thank you for bringing up this inquiry. However, we humbly point out that **Eqn. 5 is not a low-pass smoothing objective, nor was it referenced as that within our paper**. Actually, as illustrated in Sec 5, Eqn. 6 aims to adaptively smooth node features for different pairs and it enforces smoothing effect on arbitrary node pairs (instead of neighboring nodes). In this sense, the model is not a traditional low-pass filter (as GCN), and can adaptively leverage high-order information.
>
> > **W5: Mini-batch training for large graphs**
>
> The sampling technique is a common practice for training on large graphs, and adopted by peer models as it is prohibitive to fit a large graph into a GPU at the first place. While other models require sophisticated sampling, our used random mini-batch is much simpler and requires almost no extra time. Compared to NodeFormer that also uses mini-batch training on large graphs, SGFormer costs 3x fewer training time on ogbn-products, which suggests **our efficiency improvement purely stems from the simplified architecture with one-layer attention**. The latter are indeed our contributions.
>
> We will add the suggested reference in revision. Please let us know if you had further questions.

---

> > ### Comment · Reviewer_bWXq · 2023-08-13
> > **Response to Rebuttal**
> >
> > Thanks for the clarifications and it helps my understanding of the proposed method and claims.
> >
> > Some of my main concerns still remain:
> > 1) Q1 & W2: Optimal baseline results: While [1] may not have reported results the paper, benchmarks [2] using code have shown good results with these transformer(+GNN) based method. The results are better than that reported by current work and while this paper need not compare directly with it but I would at least expect a discussion as it is similar to the current work being presented. For SIGN [3] I agree it was an oversight from my side and I apologize for the same.
> > 2) Q2 & W3: Positional embeddings (PE) and role of GNNs: Without going into classifications of absolute, relative, structural etc., when we talk of PEs, it broadly refers to encodings that can help with understanding of the position or structure. Since GNNs do this on their own they do not need PEs; however transformers are not permutation invariant to the node orderings and as such need some encoding to inform of the structure or position within the graph. When the authors claim that they have proposed a transformer that doesn't need PE, it is actually the GNN that induces this invariance property. This becomes more important since the results deteriorate without GNNs and so it is an important component of the architecture and the transformer or the proposed attention mechanism in itself is not capable of learning position encoding inherently.
> > 3) Q2 & W4: Performance degrades without GNNs: I understand that the combination of the GNN with the proposed attention, which is the transformers attention without softmax, gives good results on the experimented datasets; however my concern on the contribution is that this could very well be done in the existing frameworks eg: GraphGPS, by removing the softmax and so I respectfully feel the work has limited novelty for the conference.
> >
> > I will think through the discussed points in detail and consider my review in light of the responses. \
> > Many Thanks
> >
> > [1] Masked Label Prediction: Unified Message Passing Model for Semi-Supervised Classification \
> > [2] https://paperswithcode.com/sota/node-property-prediction-on-ogbn-papers100m \
> > [3] SIGN: Scalable Inception Graph Neural Networks
> >
> > Edit: As pointed by authors, transformers are permutation invariant to node orderings like GNNs but not structure variant unlike GNNs and so need some encoding to inform of the graph structure.

---

> > > ### Author Response · Authors · 2023-08-13
> > > **Response by Authors**
> > >
> > > Thank you for raising the discussion. We next supplement more clarification to resolve your lingering concerns.
> > >
> > > **Concern 1**
> > >
> > > The models appeared on the OGB leaderboards are not commonly used for comparison in the literature, since these models rely on extra tricks (e.g., model ensembles, label reuses, data augmentation, etc.) beyond the model architectures that are our research focus. For fair comparison, our considered competitors belong to pure MLP/GNN/Transformer models trained with the standard supervised loss. The tricks used by other models on the leaderboard are orthogonal to our work as well as the competitors we compared with. We'd like to add these discussions to our Sec 5 to broaden our applicability for practitioners from the engineering side.
> > >
> > > That being said, notice that **we did not claim "SOTA performance" on particular datasets, nor was it pursued by the current work, where our main contribution is that we identify the one-layer linear attention equipped with simple GNN can achieve highly competitive results over other complicated designs**. This finding opens the possibility of Transformer-like models for handling extremely large graphs, where the scalability is a concerning bottleneck.
> > >
> > > **Concern 2**
> > >
> > > While we understand how to classify the related concepts is to a large extent a matter of semantics, we have to put these concepts in a clear way because it impacts how our claim is interpreted.
> > >
> > > The reviewer defines PEs in a broad sense that include both the positional and structural information, though precisely speaking, the positional and structural encodings refer to different aspects, since the former is often permutation-variant to input orderings (e.g., words in a sentence) and the latter is not (structures purely depend on graph properties) [1,2,3]. **No matter how PEs are defined in a broad or narrow sense, GNNs are permutation-invariant to node orderings and not recognized as PEs in the literature [1,2,3,4,5].** Our claim therefore is properly positioned with existing works and resonated by other reviewers.
> > >
> > > Transformers w/o PEs are permutation invariant to node orderings, instead of "not permutation invariant" described by the reviewer. One simple way to comprehend this is to consider the attention as a special GNN defined over a dense graph, where the latter is invariant to node orderings. Thereby, our attention already satisfies invariance propery, and the GNN is not for "inducing the invariance" but structural information that cannot be fully captured by the global attention from node features.
> > >
> > > [1] GRAPH NEURAL NETWORKS WITH LEARNABLE STRUCTURAL AND POSITIONAL REPRESENTATIONS, ICLR 2022
> > >
> > > [2] On Positional and Structural Node Features for Graph Neural Networks on Non-attributed Graphs, CIKM 2022
> > >
> > > [3] Representing long-range context for graph neural networks with global attention, NeurIPS 2021
> > >
> > > [4] A generalization of transformer networks to graphs, 2020
> > >
> > > [5] Do transformers really perform bad for graph representation?, NeurIPS 2021
> > >
> > > **Concern 3**
> > >
> > > We respectfully argue that our model is not an extension of GraphGPS which is actually designed for distinct tasks. See comparison below:
> > >
> > > ●*Problem Settings*: **GraphGPS is designed for graph classification on small graphs (up to hundreds of nodes), which is different from node classification in our work where the graph sizes are orders-of-magnitude larger (from thousands to billions). Given different scales, these two graph-related problems are often tackled separately [6, 7].** Empirically, GraphGPS cannot even scale to large graphs with 100K nodes [8].
> > >
> > > ●*Model Designs*: Our attention design is clearly different from GraphGPS, where the latter proposes to directly utilize existing Transformer attentions (e.g., Performer, Linformer, etc.). In specific, our attention does not use any approximation scheme (e.g., random features in Performer and low-rank approximation in Linformer) and can still achieve linear complexity. Besides, GraphGPS combines attention and GNN propagation at each model layer, while we combine the final results of attention and GNN at the output layer. Empirically, as compared in Table 1 in the uploaded PDF, our model significantly outperforms GraphGPS across four medium-sized datasets
> > >
> > > ●*Main Contributions*: Another critical difference is that our work emphasizes the efficacy of using one-layer attention that pushes the efficiency-accuracy frontier over existing graph Transformers including GraphGPS that stack deep attention layers and suffer the scalability issue. Backed up with our experiment and analysis, we believe the results contribute to novel aspects and can enlighten a new direction for future model designs on large graphs, which is under-explored.
> > >
> > > [6] Open Graph Benchmark: Datasets for Machine Learning on Graphs, NeurIPS 2020
> > >
> > > [7] NodeFormer: A Scalable Graph Structure Learning Transformer for Node Classification, NeurIPS 2022
> > >
> > > [8] GOAT: A GLOBAL TRANSFORMER ON LARGE-SCALE GRAPHS, ICML 2023

---

> > > > ### Comment · Reviewer_bWXq · 2023-08-13
> > > > **On Position / Structure Encodings**
> > > >
> > > > Thanks for the discussion. The authors are correct about permutation invariance of transformers. I should have used a term like "structure invariance" to imply that transformers do not respect the input graph structure. I'm sorry for mixing the terms; however the argument remains the same that GNNs help with understanding the structure and the proposed attention or transformer as a standalone is inherently not able to do so. From the response, I get that the authors are specifically referring to position encodings not being induced (as against inducing "structure encodings" from GNN) in the transformer. This could be made more concrete in the paper for better presentation and to avoid confusion.
> > > >
> > > > As a side note: To differentiate structural vs positional ecncodings, the authors mention that position encodings are permutation variant (vs structural encodings being permutation invariant). This may be true in cases where the node ordering itself is used to decide the PE such as in a text transformer. But for graphs consider for example the laplacian eigenvector which is categorized as a PE [1]. We can understand it by the following example: Consider 3 nodes with ordering (1,2,3) with following adjacency matrix [[0,1,0],[1,0,1],[0,1,0]] i.e. node 2 is the central node. Now if we exchange the order of nodes 2 and 3 (say in sequence to feed to the transformer), the adjacency now looks like [[0,0,1],[0,0,1],[1,1,0]].  If we find the eigenvectors of the laplacians for these two cases we get the same PE. So position encodings for graphs may also consider the structure, which is what the GNN helps in the absense of PE.
> > > >
> > > > [1] Recipe for a General, Powerful, Scalable Graph Transformer, NeurIPS 2022

---

> > > > > ### Author Response · Authors · 2023-08-14
> > > > > **Response by Authors**
> > > > >
> > > > > Thank you for the feedback. We are glad that our response resolves your misunderstandings.
> > > > >
> > > > > >"however the argument remains the same that GNNs help with understanding the structure and the proposed attention or transformer as a standalone is inherently not able to do so."
> > > > >
> > > > > Please kindly notice that as we compared in Sec 4, common graph Transformers are comprised of two key modules: a global attention (independent from graphs) and a graph-based module (GNN, PEs, or edge loss). **Using GNN in our model is not a disadvantage since one always need a component for incorporating the graph inductive bias into the attention that only utilizes the node features**. Compared to other models requiring PEs, our model uses simple GNN (a vanilla GCN) making the model much more efficienct and scalable (see Sec 4 and Sec 5.3 for comparison), which is particularly important for large graphs. It is also natural to see the pure attention (w/o GNNs) yields undesired performance on some datasets (e.g., Cora), which is equally encountered by other graph Transformers w/o PEs on these datasets. This does not mean the global attentions are not powerful enough, and instead, the fact is that the information contained in node features is not enough for learning desired mapping from x to y, and the graph information is useful. There also exist other cases where the graph information is not so useful and the pure attention can yield competitive performance (e.g., the heterophilic graph Actor).
> > > > >
> > > > > While our original presentation tacitly follows the commenly used definition for PEs, we will add more descriptions on this point in our Sec 4 in the revised paper (as we cannot make immediate modification to the paper PDF at the current stage) to avoid potential misunderstanding.
> > > > >
> > > > > As for the side note from the reviewer, thank you for the thoughtful discussion, **though it is outside the scope of our work which does not affect our contributions**. In fact, it still remains an open question how to design well-posed PEs on graphs, and the Laplacians PE is one instance (maybe not perfect enough).
> > > > >
> > > > > *In light of your feedback, it seems our previous response has resolved your concerns. And if our response has indeed addressed your concerns, we hope you will re-consider your score based on our initial rebuttal and discussions. Please let us know if you had any further concern.*

---

> > > > > > ### Author Response · Authors · 2023-08-16
> > > > > > **Thank you for discussions**
> > > > > >
> > > > > > Dear Reviewer bWXq,
> > > > > >
> > > > > > Thank you for the discussions. As the discussion phase approaching the end, we are writing this message to inquire if there is any further concern that causes your reservations on our works. For the three points of main concerns mentioned in your previous feedback, we provide a detailed response to fully address them. In your follow-up feedback, it seems you are still puzzled about the differentiation among positional/structural encodings and GNNs. As far as we know, in the context of graph learning, how to define and classify these concepts is still an open question and can be debatable for some cases. Apart from the work [1] you mentioned, there also exists literature [2-5] that defines positional and structural encodings as two related yet separate notions, as well as recent works [5-10] that clearly distinguishes GNNs from PEs. As illustrated in our previous response, we'd like to add more discussions to make the descriptions more clear and concrete on this point. Thank you for the nice suggestions.
> > > > > >
> > > > > > Anyway, we believe how to classify these concepts is not the focus of our paper and does not impact our main contributions. ***Our main contribution is that we identify using a simple Transformer architecture with one-layer attention can yield very competitive performance on large-graph benchmarks. This enlightens a new technical path for the open challenge of building Transformer models on large graphs. Our model SGFormer indeed demonstrates superior accuracy and efficiency over other graph Transformers and scales to the extremely large graph where others are infeasible to scale.*** We believe this finding along with our implementation SGFormer can benefit the graph ML community.
> > > > > >
> > > > > > [1] Recipe for a General, Powerful, Scalable Graph Transformer, NeurIPS 2022
> > > > > >
> > > > > > [2] GRAPH NEURAL NETWORKS WITH LEARNABLE STRUCTURAL AND POSITIONAL REPRESENTATIONS, ICLR 2022
> > > > > >
> > > > > > [3] On Positional and Structural Node Features for Graph Neural Networks on Non-attributed Graphs, CIKM 2022
> > > > > >
> > > > > > [4] Position-aware Graph Neural Networks, ICML 2019
> > > > > >
> > > > > > [5] Machine Learning on Graphs: A Model and Comprehensive Taxonomy, JMLR 2022
> > > > > >
> > > > > > [6] Global Self-Attention as a Replacement for Graph Convolution, KDD 2022
> > > > > >
> > > > > > [7] Representing Long-Range Context for Graph Neural Networks with Global Attention, NeurIPS 2021
> > > > > >
> > > > > > [8] Structure-Aware Transformer for Graph Representation Learning, ICML 2022
> > > > > >
> > > > > > [9] Transformer for Graphs: An Overview from Architecture Perspective, 2022
> > > > > >
> > > > > > [10] Do transformers really perform bad for graph representation?, NeurIPS 2021

---

> > > > > > > ### Comment · Reviewer_bWXq · 2023-08-19
> > > > > > >
> > > > > > > Thanks for the response. I get that the categorization of PEs is not the aim of this paper and was only meant for discussion. I am willing to increase my score after the discussions. But irrespective, I have one more concern regarding the cause of removing softmax and its impact. I would expect some analysis into why softmax is not beneficial or in which cases it works vs causes degradation of performance. I appreciate the authors' results that the linear mechanism gets good results but as a reviewer I have to think and ask: What is the impact of removing softmax? Are there any negatives or limitations or cases where removing softmax is detrimental to the task? I have added some references of workis that show limitations of softmax in transformer attention. For example: [2] shows that softmax attention + RPEs are not universal approximators and propose to add a learnable toeplitz matrix to fix this. [3] propose a cosine non-linearity instead of softmax. [5] shows that softmax attention causes graph transformers to be less expressive in spectral space. [6] proposes a softmax free transformer with linear complexity. If not a direct comparison, I would expect some discussion on these works in the paper.
> > > > > > >
> > > > > > > While I understand the authors' arguments that softmax causes an increase in complexity and the current paper shows transformer eliminating softmax coupled with a simple GNN is competitive, from a scientific perspective there are some unanswered questions that I would expect from the paper. Specifically: **1) From an expressivity standpoint, why does it make sense to eliminate the softmax (since the authors get better results by doing this on the experimented datasets) 2) What are the limitations and repurcussions of this change? (cases where this impacts negatively)**
> > > > > > >
> > > > > > >
> > > > > > > References:\
> > > > > > > [1] Castling-ViT: Compressing Self-Attention via Switching Towards Linear-Angular Attention at Vision Transformer Inference (CVPR 2023)\
> > > > > > > [2] Your Transformer May Not be as Powerful as You Expect (NeurIPS 2022)\
> > > > > > > [3] COSFORMER : RETHINKING SOFTMAX IN ATTENTION (ICLR 2022)\
> > > > > > > [4] SimA: Simple Softmax-free Attention for Vision Transformers (2022)\
> > > > > > > [5] How Expressive are Transformers in Spectral Domain for Graphs? (TMLR 2022)\
> > > > > > > [6] SOFT: Softmax-free Transformer with Linear Complexity (NeurIPS 2021)\
> > > > > > > [7] Escaping the Gradient Vanishing: Periodic Alternatives of Softmax in Attention Mechanism (2021)\

---

> > > > > > > > ### Author Response · Authors · 2023-08-19
> > > > > > > >
> > > > > > > > Thank you for the feedback and insightful questions. We agree that the analysis on the efficacy/expressivity of "not using softmax" is an interesting and promising direction, though it is orthogonal to our main focus. We'd respectfully recognize the reviewer's inquiry as the side aspect that can be included in our paper as a potential extension. See some explanations below.
> > > > > > > >
> > > > > > > > **Softmax v.s. no softmax**
> > > > > > > >
> > > > > > > > The quadratic complexity of softmax attention is a significant bottleneck on large graphs (our focus), which makes it infeasible for scaling to graphs with 10K nodes on a standard 16 GB GPU. In our problem, the primary reason of "not using softmax" stems from the efficiency/scalability consideration, instead of accuracy or effectiveness. The reviewer's suggested comparison is based on the effectiveness criterion, but the scalability failure of softmax attention on large graphs makes the direct comparison infeasible in common datasets with 10K-0.1B nodes.
> > > > > > > >
> > > > > > > > **Expressivity and potential limitations of the proposed attention**
> > > > > > > >
> > > > > > > > While analysis of the efficacy of softmax is not our main focus, we aim to study using one-layer global attention against multi-layer attentions. In our Sec 6, we have provided some theoretical insights into why the one-layer attention works and show that the one-layer model can be as expressive as the multi-layer model from the signal denoising perspective. These results are agnostic to specific attention functions, i.e., also applicable for softmax attention. As illustrated in our initial rebuttal, currently we do not observe obvious limitations of the proposed attention in our experimental datasets. Even so, we will explore more cases in the future.
> > > > > > > >
> > > > > > > > **More discussions on the references [1-7]**
> > > > > > > >
> > > > > > > > We appreciate the reviewer's suggested references that provide insightful discussions along an orthogonal direction to us. We will add a subsection of "further discussions" in our Sec 6 to discuss these works in detail. Thank you for the constructive suggestions.
> > > > > > > >
> > > > > > > > Besides, we also believe there potentially exists other extension aspects from our paper, especially given that building Transformers on large graphs is an under-explored area and requires distinct technical considerations due to the scalability bottleneck. This is also why we believe the results and methodology in our paper can enlighten a new direction and benefit the community.

---

> > > > > > > > > ### Comment · Reviewer_bWXq · 2023-08-20
> > > > > > > > >
> > > > > > > > > Dear authors,
> > > > > > > > >
> > > > > > > > > After considering the rebuttal and discussions I have increased my score to 5. Wish you all the best!

---

> > > > > > > > > > ### Author Response · Authors · 2023-08-20
> > > > > > > > > >
> > > > > > > > > > Dear Reviewer bWXq,
> > > > > > > > > >
> > > > > > > > > > We appreciate your nice feedback and insightful discussions that help us to clarify the big picture and can further broaden the impact of our work. We will incorporate the discussions into our revised paper. Thank you again for your dedicated time in reviewing our work.
> > > > > > > > > >
> > > > > > > > > > Best regards,
> > > > > > > > > >
> > > > > > > > > > Authors

---

### Official Review · Reviewer_moxf · 2023-06-27

**Soundness:** 3 good
**Presentation:** 4 excellent
**Contribution:** 3 good
**Rating:** 6
**Confidence:** 4

**Summary:**

The paper presents a novel approach called Simplified Graph Transformers (SGFormer) for large-graph representations using Transformers. SGFormer achieves competitive performance with just one-layer attention, eliminating the need for positional encodings, pre-processing, and augmented loss. The paper provides theoretical justification for the methodology and suggests potential future directions for building powerful and scalable Transformers on large graphs.

**Strengths:**

- SGFormer simplifies the design philosophy for Transformers on large graphs, making it highly efficient and scalable.
- The paper provides theoretical justification for the methodology, which can help in understanding and improving the approach.
- The authors experimented with ample datasets. The empirical results look promising.
- The writing is clan and easy to follow.

**Weaknesses:**

- The paper mainly focuses on node property prediction tasks, and it is unclear how well SGFormer performs on other graph-based tasks such as link prediction, node clustering, etc.

**Questions:**

- I'm curious why this simple design of message passing works. Without attention or relative distance encoding, how could the model learn to down-weight nodes that are far from the center node?

**Limitations:**

The authors discussed some limitations. I don't see any potential negative social impact.

---

> ### Author Rebuttal · Authors · 2023-08-08
>
> We thank the reviewer for the positive comments and we are glad that you liked our approach.
>
>
> > **(Q1): Why simple attention works? How the model learns to down-weight far-away nodes?**
>
> Thank you for proposing this insightful inquiry. The principle of our model is that the one-layer all-pair message passing can allow information flows among arbitrary node pairs and enable the model to adaptively leverage the global information. To further support this conjecture, we provide theoretical discussions in Section 6 from the signal denoising perspective.
>
> The model is not necessarily supposed to down-weight far-away nodes, depending on specific datasets with different properties. For instance, for heterophilic graphs or graphs with long-range dependence, the distant nodes could be informative and thus could be assigned with large attention scores. On the contrary, the case can be different for homophilic graphs. Generally speaking, we do not enforce any constraint on the attention learning and instead let the model flexibly learn useful inter-dependencies in a fully data-driven way. As further investigation, we provide more visualization for the node embeddings and graph/attention structures in Fig 1 in the uploaded PDF. We found the attentive graph is significantly different from the observed graph and the global attention can help to shorten the intra-class distance for better classification.
>
>
> > **(W1): Other graph-based tasks**
>
> We are happy to supplement more discussions on the potential applicability of SGFormer. As illustrated in Section 7, since our paper focuses on exploring a simple architecture for the fundamental challenge large-graph representations, we follow common practice using node property prediction benchmarks for evaluation. We use graphs with different sizes to broaden the scopes of our experiments, which suits the evaluation w.r.t. the model's scalability, and the results show the superiority of SGFormer on graphs with #node ranging from thousands to billions. This suggests the model can potentially be applied to other tasks defined over graph data with these scales, e.g., link prediction and node clustering. Having said that these tasks as well as more broadly domain-specific applications often require extra task-dependent designs which can be equally incorporated with our model as well as the competitors. We thereby leave exploration along this orthogonal direction as future works.
>
> Please let us know if you had further questions.

---

### Official Review · Reviewer_9PQr · 2023-07-01

**Soundness:** 3 good
**Presentation:** 3 good
**Contribution:** 3 good
**Rating:** 6
**Confidence:** 4

**Summary:**

This paper proposes Simplified Graph Transformers (SGFormer) that uses a one-layer attention to achieve competitive performance in node property prediction benchmarks.

**Strengths:**

1. The idea and the design of the simple global attention is novel.
2. The proposed SGFormer is efficient and scalable.

**Weaknesses:**

The analysis in section 6 ignores possible sub-nets (e.g. FFN with non-linearity in the original transformer architecture) between two global attention layers.

**Questions:**

1. Is the theorems still applicable when subnets exist between two global attention layers?
2. Why do you normalize the query and key in eq(2) with Frobenius norm?

**Limitations:**

Please refer to the weakness section

---

> ### Author Rebuttal · Authors · 2023-08-08
>
> Thank you for the valuable comments and insightful questions which can allow us to further illuminate the applicability of the theory.
>
> > **(Q1): Applicability of the theory when subnets exist**
>
> The theorems can be applied to the cases with FFN and non-linearity by some extension. For Theorem 1, the cost function will change after involving layer-wise FFN, and the non-linear activation can be seen as a proximal operator for gradient descent. In specific, the updating of one Transformer layer with the FFN (denoted as $h^{(k)}$) and non-linear activation (denoted as $\sigma$) can be written as
> $$
> \mathbf z_u^{(k)} = \sigma\left ( (1-\tau) h^{(k)} ( \mathbf z_u^{(k-1)})  + \tau \sum_{v=1}^N c_{uv}^{(k)} h^{(k)} ( \mathbf z_v^{(k-1)} ) \right ).
> $$
> We can prove the above updating equation is essentially a proximal gradient descent step for optimizing the new cost function
>
> $$
> \min_{\mathbf Z} \sum_{u} ||\mathbf z_u - h^{(k)}(\mathbf z_u^{(k-1)}) ||^2_2 + \lambda \sum_{u,v} c_{uv}^{(k)} ||\mathbf z_u - \mathbf z_v ||_2^2
> $$
>
> The proof can re-use that of Theorem 1 by starting with computing the gradient of the objective. Then there are two differences. First, in Eqn. 11, we need to evaluate the gradient at $\mathbf Z = h^{(k)}(\mathbf Z^{(k-1)})$ instead of $\mathbf Z = \mathbf Z^{(k-1)}$. Second, after the gradient descent, we can add a proximal operator to map the updated result into a feasible region:
>
> $$
> \mathbf z_u^{(k)} = Prox_{\Omega} \left ( (1-\tau) h^{(k)} ( \mathbf z_u^{(k-1)})  + \tau \sum_{v=1}^N c_{uv}^{(k)} h^{(k)} ( \mathbf z_v^{(k-1)} )   \right ),
> $$
>
> where $Prox_{\Omega} (z) = \arg\min_{x\in \Omega} d(x, z) $, $\Omega$ defines a feasible region and $d$ denotes the distance (e.g., l2). This gives rise to a proximal gradient descent which still guarantees a strict minimization for the objective, and the above equation coincides with the one-layer updating equation shown at the beginning. In particular, if one uses ReLU activation, the proximal operator will be $\mbox{Prox}_{\Omega} (z) = \max(0, z)$.
>
> For Theorem 2, if one only considers FFN without non-linearity, then the proof can be trivially extended to obtain the same result. In specific, for Eqn. 15, one can alter the order of the weight matrices and propagation matrices:
> $$
> \mathbf Z^{(K)} = \mathbf P^{(K)} \mathbf W^{(K)} \cdots \mathbf P^{(1)} \mathbf W^{(1)} \mathbf Z^{(0)} = \mathbf P^{(K)} \cdots \mathbf P^{(1)} \mathbf W^{(K)} \cdots \mathbf W^{(1)} \mathbf Z^{(0)} = \mathbf P^* \mathbf W^* \mathbf Z^{(0)}
> $$
>
> In this way, we can similarly construct a propagation matrix $\mathbf P^*$ that yields the same result as the multi-layer model. Furthermore, if we consider the FFN with non-linear activation, the proof cannot be directly adapted, though the result still loosely holds by considering the FFN as a two-layer MLP with universal approximation power. In this situation, the one-layer attention model can approximate any function mapping from $\mathbf Z^{(0)}$ to $\mathbf Z^{(K)}$ yielded by the multi-layer attentive propagation.
>
> > **(Q2): Why use normalization with Frobenius norm**
>
> The normalization can restrict the magnitude of query and key vectors and avoid the explosion of the attention scores. In practice, we found the normalization works smoothly across different datasets and can yield stably good results.
>
> Please let us know if you had further questions.

---

### Official Review · Reviewer_J3Wh · 2023-07-09

**Soundness:** 3 good
**Presentation:** 4 excellent
**Contribution:** 3 good
**Rating:** 7
**Confidence:** 4

**Summary:**

SGFormer adapts the transformer to large graphs by replacing the usual quadratic self-attention mechanism with a convex combination of linear attention and an arbitrary Graph Neural Network (GNN). The benchmark using node prediction tasks from Open Graph Benchmark (OGB) and achieve SOTA results. Moreover, the method is fast and scales to large graphs. They provide some theoretical justification why their one layer method can be as good as multiple layers.

**Strengths:**

The paper is well-written. Although dense, it provides a good summary of related works and experiments and methodology are clear.

Results are very strong in terms of model quality and performance.

The simplicity of the method is appreciated.

**Weaknesses:**

There seems to be a lot of hyperparameters in the attention mechanism like \alpha and \beta that were not ablated? Also the structure of the GNN may matter a lot?

**Questions:**

I would be curious if there is any way to figure out the contribution of the GNN and linear attention? Is there a way to get some intuition like looking at attention scores?

Maybe some clustering of the embedding space could be interesting.

**Limitations:**

The paper only focuses on node prediction tasks.

---

> ### Author Rebuttal · Authors · 2023-08-08
>
> Thank you for the positive comments and constructive suggestions that can help to further improve our work.
>
> > **(W1): Discussions of $\alpha$, $\beta$ and importance of graphs**
>
> In Table 6 (in the appendix), we provide ablation studies for different values of $\alpha$ (the weight of GNN). The results show that the GNN module is particularly important for homophily graphs (like Cora) and less important for heterophily graphs (like Actor). Since the GNN module can utilize the input graph information complementary to the all-pair global attention, its importance would depend on the informativeness of graph structures in different cases.
>
> Besides, we run some new experiments on CiteSeer and Chameleon with different settings of $\beta$, and report the results in Table 3 in the uploaded PDF. We found the model is overall insensitive to $\beta$ within a proper range.
>
> > **(Q1): Visualization of attention and node embeddings**
>
> We originally present some visualization for the attention scores in Fig. 16 in the appendix. We found that the attention matrices (that have some dominant lines) tend to have different patterns with the graph adjacency matrices (that are sparse and scattered). This suggests the attention indeed learns some useful information from data beyond the input graph.
>
> Also, following the advice, we add some visualization of the node embeddings in Fig 1 in the uploaded PDF. We found the attentive graph (by filtering the edges with attention weights larger than a threshold) is significantly different from the observed graph. Besides, the global attention of SGFormer pushes the node embeddings to have smaller intra-class distance than those of GCN.
>
> Please let us know if you had further questions.

---

### Author Rebuttal · Authors · 2023-08-08

Dear Area Chairs and Reviewers,

We thank the reviewers for their time, thorough reviews and constructive suggestions. Overall, the reviewers deemed our work well written, the method novel and the results solid. Below we first restate our main contributions and distill the reviewers' comments to facilitate the discussions.

**1. We reassess the need for deep attention layers when learning representations on large graphs, and propose Simplified Graph Transformers that unlock the potential of one-layer global attention for large graphs where the scalability matters.**

>Most of the reviewers appreciated our motivation/methodology with the comments (R**J3Wh**:"the simplicity of the method is appreciated", R**9PQr**:"the idea and the design of the simple global attention is novel", R**moxf**:"SGFormer simplifies the design philosophy for Transformers on large graphs").

**2. Despite the simplicity, our model shows highly competitive results across diverse node property prediction benchmarks and its superior scalability to graph sizes ranging from 2K to 0.1B, and yields up to 37x/141x speedup for training/inference over efficient graph Transformers.**

> Most of the reviewers recognized our results significant with the comments (R**J3Wh**:"Results are very strong in terms of model quality and performance", R**9PQr**:"The proposed SGFormer is efficient and scalable", R**moxf**:"The empirical results look promising")

**3. As a justification, we provide theoretical insights into why the one-layer attention model can be a powerful graph learner through the lens of signal denoising perspectives.**

> R**moxf** acknowledged that our analysis "can help in understanding and improving the approach". R**9PQr** inquires the potential extension of our results and we show how to generalize the analysis in the individual response.

While R**bWXq** raised questions on our comparison results and asked for more empirical justifications, we identify some factual errors in the quoted scores and concept descriptions which may have led to the reviewer's misinterpretations regarding our claims and results. We clarify and provide detailed elaboration in the separate rebuttal.

We upload a one-page PDF that contains new experimental results as suggested by the reviewers. In the following individual response, we provide answers to each raised weakness/question points and will incorporate the suggestions and new results in our revised paper.

Best regards,

Authors of Submission 11333

---

### Author Response · Authors · 2023-08-18

Dear Respected Reviewers,

We'd like to express our sincerest gratitude for your dedicated time and meticulous review of our paper. Your insightful comments and invaluable suggestions have played a pivotal role in refining the quality of our work. We have thoroughly considered your concerns and carefully addressed each of your questions. As the rebuttal phase unfolds, we eagerly await your post-rebuttal feedback. We would appreciate your feedback on whether our responses have satisfactorily resolved your concerns.

With warm regards,

Authors

---

### Decision · Program_Chairs · 2023-09-21

**Decision:**

Accept (poster)

**Comment:**

This paper presents the Simplified Graph Transformers (SGFormer) for graph representations learning. The method greatly simplifies the training process in prior work and achieves good performance on several tasks. Since all reviewers are positive, I am happy to accept the paper.